# Theoretically Better and Numerically Faster Distributed Optimization with Smoothness-Aware Quantization Techniques

**Bokun Wang**
Texas A&M University, United States*
bokunw.wang@gmail.com

**Mher Safaryan**
KAUST, Saudi Arabia
mher.safaryan.1@kaust.edu.sa

**Peter Richtárik**
KAUST, Saudi Arabia
peter.richtarik@kaust.edu.sa

## Abstract

To address the high communication costs of distributed machine learning, a large body of work has been devoted in recent years to designing various compression strategies, such as sparsification and quantization, and optimization algorithms capable of using them. Recently, Safaryan et al. [2021] pioneered a dramatically different compression design approach: they first use the local training data to form local *smoothness matrices* and then propose to design a compressor capable of exploiting the smoothness information contained therein. While this novel approach leads to substantial savings in communication, it is limited to sparsification as it crucially depends on the linearity of the compression operator. In this work, we generalize their smoothness-aware compression strategy to *arbitrary unbiased compression* operators, which also include sparsification. Specializing our results to stochastic quantization, we guarantee significant savings in communication complexity compared to standard quantization. In particular, we prove that block quantization with $n$ blocks theoretically outperforms single block quantization, leading to a reduction in communication complexity by an $\mathcal{O}(n)$ factor, where $n$ is the number of nodes in the distributed system. Finally, we provide extensive numerical evidence with convex optimization problems that our smoothness-aware quantization strategies outperform existing quantization schemes as well as the aforementioned smoothness-aware sparsification strategies with respect to three evaluation metrics: the number of iterations, the total amount of bits communicated, and wall-clock time.

## 1 Introduction

Training modern machine learning models is typically cast in terms of (regularized) empirical risk minimization problem and requires increasingly more training data to make empirical risk closer to the true risk [Schmidhuber, 2015, Vaswani et al., 2019]. This natural requirement makes it harder (and in some scenarios impossible) to collect all data in one place and carry out the training using a single data source. As a result, we reconciled with a flock of datasets disseminated across various compute nodes holding the actual training data [Bekkerman et al., 2011, Vogels et al., 2019]. However, such divide-and-conquer approach of handling vast amount of data means that local updates need to be communicated among the nodes (or through some central server orchestrating the process), which

---

*Work done when the author was a research intern at KAUST, Saudi Arabia.

36th Conference on Neural Information Processing Systems (NeurIPS 2022).

often forms the main bottleneck in modern distributed systems [Zhang et al., 2017, Lin et al., 2018]. This issue is further exacerbated by the fact that modern highly performing models are typically overparameterized [Brown et al., 2020, Narayanan et al., 2021].

**1.1. Distributed training.** In this paper, we consider distributed training formalized as the following optimization problem

$$\min_{x \in \mathbb{R}^d} f(x) + R(x), \quad \text{where} \quad f(x) \stackrel{\text{def}}{=} \frac{1}{n} \sum_{i=1}^{n} f_i(x), \tag{1}$$

and where $d$ is the number of parameters of model $x \in \mathbb{R}^d$ to be trained, $n$ is the number of machines/nodes participating in the training, $f_i(x)$ is the loss/risk associated with the data stored on machine $i \in [n] \stackrel{\text{def}}{=} \{1, 2, \ldots, n\}$, $f(x)$ is the empirical loss/risk, and $R(x)$ is a regularizer.

Because of the communication constraints, large body of work has been devoted in recent years to the design of various compression strategies, such as sparsification [Konečný and Richtárik, 2018, Wangni et al., 2018, Alistarh et al., 2018], quantization [Goodall, 1951, Roberts, 1962, Alistarh et al., 2017], low-rank approximation [Vogels et al., 2019], three point compressor [Richtárik et al., 2022], and optimization algorithms capable of using them, such as Distributed Compressed Gradient Descent (DCGD) [Khirirat et al., 2018], QSGD [Alistarh et al., 2017, Faghri et al., 2020], NUQSGD [Ramezani-Kebrya et al., 2021], DIANA [Mishchenko et al., 2019, Horváth et al., 2019], PowerSGD [Vogels et al., 2019], signSGD [Bernstein et al., 2018, Safaryan and Richtárik, 2021], intSGD [Mishchenko et al., 2021], ADIANA [Li et al., 2020], MARINA [Gorbunov et al., 2021].

**1.2. From scalar smoothness to matrix smoothness.** Typically, distributed optimization algorithms in the literature that employ compressed communication, including all methods from the aforementioned works, use only shallow smoothness information of the loss function such as scalar $L$-smoothness [Nesterov, 2004].

**Definition 1** (Scalar Smoothness)**.** Differentiable function $\phi : \mathbb{R}^d \to \mathbb{R}$ is called $L$-smooth if there exists a non-negative scalar value $L \geq 0$ such that

$$\phi(x) \leq \phi(y) + \langle \nabla \phi(y), x - y \rangle + \frac{L}{2} \|x - y\|^2, \quad \forall x, y \in \mathbb{R}^d. \tag{2}$$

As pointed out by Safaryan et al. [2021], smoothness constant $L$ reflects small part of the rich smoothness information often easily available through the training data. In their recent work, Safaryan et al. [2021] pioneered a dramatically different compression design approach. First, they propose to use the local training data to form local *smoothness matrices*, which they claim contain much more useful information than standard smoothness constants.

**Definition 2** (Matrix Smoothness)**.** Differentiable function $\phi : \mathbb{R}^d \to \mathbb{R}$ is called $\mathbf{L}$-smooth if there exists a symmetric positive semidefinite matrix $\mathbf{L} \succeq \mathbf{0}$ such that

$$\phi(x) \leq \phi(y) + \langle \nabla \phi(y), x - y \rangle + \frac{1}{2} \|x - y\|_{\mathbf{L}}^2, \quad \forall x, y \in \mathbb{R}^d. \tag{3}$$

Intuitively, the usefulness of $\mathbf{L}$-smoothness over the standard $L$-smoothness is the tighter upper bound for functional growth. In other words, if for a function $\phi$ we have the tightest scalar smoothness $L$ and the tightest matrix smoothness $\mathbf{L}$ parameters, then $L = \lambda_{\max}(\mathbf{L})$ and, hence, upper bound (3) is better than (2). To understand the relationship deeper, consider the functional growth of $\phi$ along the direction $e \in \mathbb{R}^d$ (without loss of generality assume $\|e\| = 1$). Let $x = y + te$, where $t > 0$ is a positive scaling parameter, and consider the quadratic terms $\frac{t^2}{2} L$ of (2) and $\frac{t^2}{2} \langle e, \mathbf{L}e \rangle$ of (3) bounded the functional growth. Obviously, depending on the direction $e$, the quadratic form $\langle e, \mathbf{L}e \rangle$ can be much smaller than $L = \lambda_{\max}(\mathbf{L}) = \sup\{\langle e, \mathbf{L}e \rangle : \|e\| = 1\}$.

Non-uniform functional growths over different directions hints to design optimization algorithms that are aware of such properties of the objective. Several works on randomized coordinate descent have successfully exploited this approach [Richtárik and Takáč, 2016, Qu and Richtárik, 2016a,b, Hanzely and Richtárik, 2019, Hanzely and Richtárik, 2019]. For example, the 'NSync algorithm of Richtárik and Takáč [2016] uses the smoothness matrix to estimate smaller, so-called *ESO (Expected Separable Overapproximation)* parameters for each coordinate, leading to larger stepsizes for the update rule and improved complexity for the algorithm. Note that randomized coordinate descent can be viewed as compressed gradient descent with random sparsification ($n = 1$ number of workers).

In the context of distributed optimization, using smoothness matrices $\mathbf{L}_i$ of all local loss functions $f_i(x), \ i \in [n]$, Safaryan et al. [2021] design a compressor capable of exploiting the smoothness

**Table 1:** Summary of main theoretical results of this work. Below constants and $\log \frac{1}{\varepsilon}$ factors are hidden, $n$ is the number of nodes, $d$ is the model size, $L_{\max} = \max_i L_i$, $L_i = \lambda_{\max}(\mathbf{L}_i)$, the expected smoothness constant $\mathcal{L}_{\max}$ is defined in (4), the variance of generic compression operator is denoted by $\omega$, parameters $\nu$ and $\nu_1$ are defined in (8). See Table 3 in the Appendix for the notations. We discuss some limitations of the proposed algorithms in Section A.3.

| Regime | $\nabla f_i(x^*) \equiv 0$ | arbitrary $\nabla f_i(x^*)$ |
|---|---|---|
| **Original Methods** | **DCGD [Khirirat et al., 2018]** | **DIANA [Mishchenko et al., 2019]** |
| Iteration Complexity | $\frac{L}{\mu} + \frac{\omega L_{\max}}{n\mu}$ | $\omega + \frac{L_{\max}}{\mu} + \frac{\omega L_{\max}}{n\mu}$ |
| Communication Complexity Standard Quantization ($\omega = \mathcal{O}(n)$) | $d\frac{L_{\max}}{\mu}$ | $nd + d\frac{L_{\max}}{\mu}$ |
| **Redesigned Methods** | **DCGD+ (Algorithm 1)** **with general compression** | **DIANA+ (Algorithm 2)** **with general compression** |
| Iteration Complexity | $\frac{L}{\mu} + \frac{\mathcal{L}_{\max}}{n\mu}$ | $\omega_{\max} + \frac{L}{\mu} + \frac{\mathcal{L}_{\max}}{n\mu}$ |
| Communication Complexity Block Quantization ($n = \mathcal{O}(\sqrt{d})$) | $\frac{d}{n}\frac{L_{\max}}{\mu}$ (if $\nu$, $\nu_1$ are $\mathcal{O}(1)$) | $nd + \frac{d}{\sqrt{nd}}\frac{L_{\max}}{\mu}$ (if $\nu$, $\nu_1$ are $\mathcal{O}(1)$) |
| Communication Complexity Quantization with varying steps | $\frac{d}{n}\frac{L_{\max}}{\mu} + \frac{d}{d}\frac{L_{\max}}{\mu}$ (if $\nu$, $\nu_1$ are $\mathcal{O}(1)$) | $nd + \frac{d}{n}\frac{L_{\max}}{\mu} + \frac{d}{d}\frac{L_{\max}}{\mu}$ (if $\nu$, $\nu_1$ are $\mathcal{O}(1)$) |
| Theorems | 1, 3, 5 | 2, 4, 6 |
| Speedup factor (up to) | $\min(n, d)$ | $\min(n, d)$ |

information contained within the smoothness matrices. In particular, under certain heterogeneity conditions on the smoothness matrices $\mathbf{L}_i$, their new compressor reduces total communication cost by a factor of $\mathcal{O}(\min(n, d))$.

*While this novel approach leads to substantial savings in communication, it is limited to random sparsification as it crucially depends on the linearity of the compression operator. It is not clear whether this approach can be useful in the design of other smoothness-aware compression techniques.*

## 2 Summary of Contributions

Motivated by the above mentioned development, in this work, we made the following contributions.

**2.1. Extending matrix-smoothness-aware sparsification to general compression schemes.** First, we generalize the smoothness-aware sparsification strategy [Safaryan et al., 2021] to arbitrary unbiased compressors. Instead of sparsification operator, we consider the generic class $\mathbb{B}^d(\omega)$ of (possibly randomized) unbiased compression operators $\mathcal{C} \colon \mathbb{R}^d \to \mathbb{R}^d$ with bounded variance $\omega \geq 0$, i.e.,

$$\mathbb{E}[\mathcal{C}(x)] = x, \quad \mathbb{E}\left[\|\mathcal{C}(x) - x\|^2\right] \leq \omega \|x\|^2, \quad \forall x \in \mathbb{R}^d.$$

This class is quite broad including random sparsification and various quantization schemes. To benefit from the matrix smoothness information with general compressor $\mathcal{C}$, we propose the following modification in the communication protocol. If $x \in \mathbb{R}^d$ is the vector to be communicated, instead of applying compressor $\mathcal{C}$ directly to $x$ and sending $\mathcal{C}(x)$, we compress it by $\mathcal{C}(\mathbf{L}^{\dagger 1/2}x)$ and decompress it by multiplying $\mathbf{L}^{1/2}$. Overall, the receiver estimates the original $x$ by $\mathbf{L}^{1/2}\mathcal{C}(\mathbf{L}^{\dagger 1/2}x)$.

**2.2. Distributed compressed methods with improved communication complexity.** To highlight the appropriateness of our generalization, we redesign two distributed compressed methods—DCGD [Khirirat et al., 2018] and DIANA [Mishchenko et al., 2019]—to effectively utilize both matrix smoothness information and general compression operators leading to new methods, which we call DCGD+ (Algorithm 1) and DIANA+ (Algorithm 2). The key notion we introduce that enables the technical analysis is the following quantity describing interaction between compression operator $\mathcal{C} \in \mathbb{B}^d(\omega)$ and smoothness matrix $\mathbf{L} \succeq \mathbf{0}$:

$$\mathcal{L}(\mathcal{C}, \mathbf{L}) \stackrel{\text{def}}{=} \inf\left\{\mathcal{L} \geq 0 \colon \mathbb{E}\|\mathcal{C}(x) - x\|_{\mathbf{L}}^2 \leq \mathcal{L}\|x\|^2\right\} \leq \omega \lambda_{\max}(\mathbf{L}).$$

This quantity generalizes the one defined in Safaryan et al. [2021] for sparsification, and provides means for tighter theoretical guarantees (Theorems 1 and 2) and better compression design.

**2.3. Block quantization.** As we are no longer constrained to sparsification to exploit matrix smoothness, we consider more aggressive quantization schemes to further reduce the communication cost. Our first extension of standard quantization [Alistarh et al., 2017] is *block quantization*, where each block is allowed to have a separate quantization parameter. Notably, we show theoretically that our block quantization with $n$ blocks outperforms single block quantization and saves in communication by a factor of $\mathcal{O}(n)$ for both DCGD+ (Theorem 3) and DIANA+ (Theorem 4) when $n = \mathcal{O}(\sqrt{d})$.

**2.3. Quantization with varying steps.** In our second extension of standard quantization, we go even further and allow all coordinates to have their own quantization steps. This extension turns out to be more efficient in practice than block quantization and provides savings in communication cost by a factor of $\mathcal{O}(\min(n, d))$ for both DCGD+ (Theorem 5) and DIANA+ (Theorem 6).

**2.4. Experiments.** Finally, we perform extensive numerical experiments using LibSVM data [Chang and Lin, 2011] and provide clear numerical evidence that the proposed smoothness-aware quantization strategies outperform existing quantization schemes as well the aforementioned smoothness-aware sparsification strategies with respect to three evaluation metrics: the number of iterations, the total amount of bits communicated, and wall-clock time (see Section 6 and the Appendix).

## 3  Smoothness-Aware Distributed Methods with General Compressors

In this section we extend methods DCGD+ and DIANA+ of Safaryan et al. [2021] to handle arbitrary unbiased compression operators. We consider the problem (1) with matrix smoothness assumption for all local losses $f_i(x)$ and with strong convexity of loss function $f(x)$.

**Assumption 1** (Matrix smoothness). The functions $f_i \colon \mathbb{R}^d \to \mathbb{R}$ are differentiable, convex, lower bounded and $\mathbf{L}_i$-smooth. Besides, $f$ is $\mathbf{L}$-smooth with the scalar smoothness constant $L \overset{\text{def}}{=} \lambda_{\max}(\mathbf{L})$.

First, note that lower boundedness of $f_i(x)$ is not needed once $\mathbf{L}_i \succ 0$ is invertible. This part of the assumption is not a restriction in applications as all loss function are lower bounded. Regarding the relation between $\mathbf{L}$ and $\mathbf{L}_i$, notice that (1) implies $\mathbf{L} \preceq \frac{1}{n} \sum_{i=1}^n \mathbf{L}_i$. This means that while $\frac{1}{n} \sum_{i=1}^n \mathbf{L}_i$ can serve as a smoothness matrix for $f$, there might be a tighter estimate, which we denote by $\mathbf{L}$. Clearly, matrix smoothness provides much more information about the loss function than scalars smoothness. However, estimating dense smoothness matrix $\mathbf{L}$ could be expensive for problems beyond generalized linear models because of the $d^2$ number of entries and lack of closed-form expression. On the other hand, estimating sparse, such as diagonal, smoothness matrix $\mathbf{Diag}(L^1, L^2, \ldots, L^d)$ should be feasible. Lastly, if $\mathbf{L}$ is a smoothness matrix (could be dense, diagonal or any structure) of $f$, then any matrix $\widetilde{\mathbf{L}} \preceq \mathbf{L}$ is also a smoothness matrix for $f$. This implies that our theory would still work if the smoothness matrix is over-approximated.

**Assumption 2** ($\mu$-convexity). The function $f \colon \mathbb{R}^d \to \mathbb{R}$ is $\mu$-convex for some $\mu > 0$, i.e.,

$$f(x) \geq f(y) + \langle \nabla f(y), x - y \rangle + \tfrac{\mu}{2} \|x - y\|^2, \quad \forall x, y \in \mathbb{R}^d.$$

This assumption is rather standard in the literature, sometimes referred to as strong convexity.

**3.1. DCGD+ with arbitrary unbiased compression.** In our version of DCGD+, each node $i \in [n]$ is allowed to control its own compression operator $\mathcal{C}_i \in \mathbb{B}^d(\omega)$ independent of other nodes. Denote

$$\mathcal{L}_{\max} \overset{\text{def}}{=} \max_{1 \leq i \leq n} \mathcal{L}_i, \quad \text{where} \quad \mathcal{L}_i \overset{\text{def}}{=} \mathcal{L}(\mathcal{C}_i, \mathbf{L}_i). \tag{4}$$

Furthermore, as the compressor $\mathcal{C}_i$ can be random, denote by $\mathcal{C}_i^k$ a copy of $\mathcal{C}_i$ generated at iteration $k$.

---

**Algorithm 1** DCGD+ WITH ARBITRARY UNBIASED COMPRESSION

1: **Input:** Initial point $x^0 \in \mathbb{R}^d$, step size $\gamma > 0$, compression operators $\{\mathcal{C}_1^k, \ldots, \mathcal{C}_n^k\}$
2: **on** server
3:     send $x^k$ to all nodes
4:     get compressed updates $\mathcal{C}_i^k(\mathbf{L}_i^{\dagger 1/2} \nabla f_i(x^k))$ from all nodes $i \in [n]$
5:     update the model to $x^{k+1} = \operatorname{prox}_{\gamma R}(x^k - \gamma g^k)$, where $g^k = \frac{1}{n} \sum_{i=1}^n \mathbf{L}_i^{1/2} \mathcal{C}_i^k(\mathbf{L}_i^{\dagger 1/2} \nabla f_i(x^k))$

---

Similar to the standard DCGD method, convergence of DCGD+ is linear up to some oscillation neighborhood. However, for the interpolation regime this neighborhood vanishes and the method converges linearly to the exact solution.

**Theorem 1.** *Let Assumptions 1 and 2 hold and assume that each node $i \in [n]$ generates its own copy of compression operator $\mathcal{C}_i^k \in \mathbb{B}^d(\omega_i)$ independently from others. Then, for the step-size $0 < \gamma \leq \frac{1}{L + \frac{2}{n}\mathcal{L}_{\max}}$, the iterates $\{x^k\}$ of DCGD+ (Algorithm 1) satisfy*

$$\mathbb{E}\left[\|x^k - x^*\|^2\right] \leq (1 - \gamma\mu)^k \|x^0 - x^*\|^2 + \frac{2\gamma\sigma_+^*}{\mu n}, \tag{5}$$

*where $\sigma_+^* \overset{\text{def}}{=} \frac{1}{n}\sum_{i=1}^n \mathcal{L}_i\|\nabla f_i(x^*)\|_{\mathbf{L}_i^\dagger}^2$. In particular, for the interpolation regime (i.e., $\nabla f_i(x^*) = 0$ for all $i \in [n]$), then DCGD+ converges linearly with iteration complexity*

$$\mathcal{O}\left(\left(\frac{L}{\mu} + \frac{\mathcal{L}_{\max}}{n\mu}\right)\log\frac{1}{\varepsilon}\right). \tag{6}$$

We show later that the iteration complexity (6) of DCGD+ can be much better than one of DCGD. However, the size of the neighborhood of DCGD+ might be bigger than of DCGD. In case of standard (scalar) smoothness (i.e. $\mathbf{L}_i = L_i\mathbf{I}$) the size of the neighborhood would be $\sigma^* \overset{\text{def}}{=} \frac{1}{n}\sum_{i=1}^n \omega_i\|\nabla f_i(x^*)\|^2$, which might be smaller than $\sigma_+^*$. Even though we have $\mathcal{L}_i \leq \omega_i\lambda_{\max}(\mathbf{L}_i)$ from the definition of $\mathcal{L}_i$, it does not imply $\mathcal{L}_i\mathbf{L}_i^\dagger \preceq \omega_i\mathbf{I}$. Thus, with matrix-smoothness-aware compression we ensure faster linear convergence at the cost of a possibly larger oscillation radius. This is not an issue for the interpolation regime, which can interpolate the whole training data with zero loss. Moreover, next we present an algorithmic solution to remove the neighborhood using the DIANA method.

**3.2. DIANA+ with arbitrary unbiased compression.** The mechanism allowing to remove the neighborhood in DIANA+ is based on the DIANA method, which was initially introduced for ternary quantization by Mishchenko et al. [2019], and then extended to arbitrary unbiased compression operators by Horváth et al. [2019]. The high level idea is to learn the local optimal gradients $\nabla f_i(x^*)$ by estimates $u_i^k$ for all nodes $i \in [n]$ in a communication efficient manner. Nodes use these estimates $u_i^k$ to progressively construct better local gradient estimates $g_i^k$ reducing the variance induced from the compression.

---

**Algorithm 2** DIANA+ WITH ARBITRARY UNBIASED COMPRESSION

---

1: **Input:** Initial point $x^0 \in \mathbb{R}^d$, initial shifts $u_i^0 \in \text{range}(\mathbf{L}_i)$ and $u^0 \overset{\text{def}}{=} \frac{1}{n}\sum_{i=1}^n u_i^0$, step size parameters $\gamma > 0$ and $\alpha > 0$, compression operators $\{\mathcal{C}_1^k, \ldots, \mathcal{C}_n^k\}$
2: **for** each node $i = 1, \ldots, n$ in parallel **do**
3:     get $x^k$ from the server and compute local gradient $\nabla f_i(x^k)$
4:     send compressed update $\Delta_i^k = \mathcal{C}_i^k(\mathbf{L}_i^{\dagger 1/2}(\nabla f_i(x^k) - u_i^k))$ to the server
5:     update local gradient and shift $\overline{\Delta}_i^k = \mathbf{L}_i^{1/2}\Delta_i^k$, $g_i^k = u_i^k + \overline{\Delta}_i^k$, $u_i^{k+1} = u_i^k + \alpha\overline{\Delta}_i^k$
6: **end for**
7: **on** server
8:     get all sparse updates $\Delta_i^k$, $i \in [n]$ and $\overline{\Delta}^k = \frac{1}{n}\sum_{i=1}^n \overline{\Delta}_i^k = \frac{1}{n}\sum_{i=1}^n \mathbf{L}_i^{1/2}\Delta_i^k$, $g^k = \overline{\Delta}^k + u^k$
9:     update the global model to $x^{k+1} = \text{prox}_{\gamma R}(x^k - \gamma g^k)$ and global shift to $u^{k+1} = u^k + \alpha\overline{\Delta}^k$

---

We prove in the Appendix that both iterates $x^k$ and all local gradient estimates $u_i^k$ converge linearly to the exact solution $x^*$ and $\nabla f_i(x^*)$ respectively.

**Theorem 2.** *Let Assumptions 1 and 2 hold and assume that each node $i \in [n]$ generates its own copy of compression operator $\mathcal{C}_i^k \in \mathbb{B}^d(\omega_i)$ independently from others. Then, if $\omega_{\max} = \max_{1 \leq i \leq n} \omega_i$ and the step-size $\gamma = \frac{1}{L + \frac{6}{n}\mathcal{L}_{\max}}$, DIANA+ (Algorithm 2) converges linearly with iteration complexity*

$$\mathcal{O}\left(\left(\omega_{\max} + \frac{L}{\mu} + \frac{\mathcal{L}_{\max}}{n\mu}\right)\log\frac{1}{\varepsilon}\right). \tag{7}$$

Notice that the cost of removing the neighborhood is the extra $\mathcal{O}(\omega_{\max}\log\frac{1}{\varepsilon})$ iterations, which is negligible in the overall complexity (7) above. Another interesting observation is the second order

flavor of the gradient learning technique employed by DIANA+. Let, for concreteness, matrices $\mathbf{L}_i$ be invertible and $\mathcal{C}_i^k(-x) = -\mathcal{C}_i^k(x)$ for all $x \in \mathbb{R}^d$ (both random sparsification and quantization satisfy this). Typically, the learning procedure of the original DIANA method, $u_i^{k+1} = u_i^k - \alpha \mathcal{C}_i^k(u_i^k - \nabla f_i(x^k))$, can be interpreted as a single step of CGD applied to the problem of minimizing the convex quadratic function $\varphi_i^k(u) \stackrel{\text{def}}{=} \frac{1}{2} \left\| u - \nabla f_i(x^k) \right\|^2$, which changes in each iteration because the gradient changes. In contrast, we observe that the learning mechanism of DIANA+ can be interpreted as a single step of a (damped) Newton's method with compressed gradients and with the true Hessian. Indeed, fix the iteration counter $k$ and denote $\varphi_i^k(u) \stackrel{\text{def}}{=} \frac{1}{2} \left\| u - \nabla f_i(x^k) \right\|_{\mathbf{L}_i^{-1/2}}^2$.
Then, the update rule of shifts $u_i^k$ in DIANA+ can be rewritten as $u_i^{k+1} = u_i^k - \alpha \mathbf{L}_i^{1/2} \mathcal{C}(\mathbf{L}_i^{-1/2}(u_i^k - \nabla f_i(x^k))) = u_i^k - \alpha \left[\nabla^2 \varphi_i(u_i^k)\right]^{-1} \mathcal{C}_i^k(\nabla \varphi_i(u_i^k))$. This might serve as an extra explanation on why incorporating smoothness matrices properly can improve the performance of first order methods with communication compression.

**3.3. Baselines for the original methods.** To make the theoretical comparison against DCGD and DIANA more transparent, we fix the following baselines using the standard quantization scheme.

• **Baseline for DIANA.** The iteration complexity of DIANA is $T = \widetilde{\mathcal{O}}(\omega + \frac{L_{\max}}{\mu} + \frac{\omega L_{\max}}{n\mu})$. When applying standard quantization from [Alistarh et al., 2017], the amount of bits each node communicates is $b = \mathcal{O}(s^2 + s\sqrt{d}) = \max(s^2, s\sqrt{d}) = \mathcal{O}(\frac{d}{\omega})$ since $\omega = \min(\frac{d}{s^2}, \frac{\sqrt{d}}{s}) = \frac{d}{\max(s^2, s\sqrt{d})}$ (Lemma 3.1., Alistarh et al. [2017]). Thus, the total communication complexity of DIANA is $n \cdot T \cdot b = \widetilde{\mathcal{O}}(nd + \frac{ndL_{\max}}{\omega\mu} + \frac{dL_{\max}}{\mu})$. Thus, the optimal total communication complexity of DIANA is $\widetilde{\mathcal{O}}(nd + \frac{dL_{\max}}{\mu})$, which is attained when $\omega = \mathcal{O}(n)$.

• **Baseline for DCGD.** Based on the iteration complexity[2] $\widetilde{\mathcal{O}}(\frac{L}{\mu} + \frac{\omega L_{\max}}{n\mu})$ of DCGD (in case $\nabla f_i(x^*) = 0$ for all $i \in [n]$), we fix the same level of compression $\omega = \mathcal{O}(n)$, which results in $\widetilde{\mathcal{O}}(\frac{L_{\max}}{\mu})$ iterations complexity. From the estimate of quantization variance $\omega = \min\left(\frac{d}{s^2}, \frac{\sqrt{d}}{s}\right)$ we conclude that $s = \mathcal{O}(\frac{\sqrt{d}}{n})$ should be used. Finally, with this choice of $s$, each node communicates $\mathcal{O}(s^2 + s\sqrt{d}) = \mathcal{O}(\frac{d}{n})$ amount of bits. Thus, total communication complexity (i.e. how many bits flows through the central server) of DCGD is $\widetilde{\mathcal{O}}(\frac{dL_{\max}}{\mu})$.

To compare the proposed methods with these baselines and highlight improvement factors, define parameters $\nu$ and $\nu_1$ describing local smoothness matrices $\mathbf{L}_i$ as follows

$$\nu \stackrel{\text{def}}{=} \frac{\sum_{i=1}^n L_i}{\max_{i\in[n]} L_i}, \quad \nu_1 \stackrel{\text{def}}{=} \max_{i\in[n]} \frac{\sum_{j=1}^d \mathbf{L}_{i;j}}{\max_{j\in[d]} \mathbf{L}_{i;j}}, \tag{8}$$

where $L_i = \lambda_{\max}(\mathbf{L}_i)$, $L_{\max} \stackrel{\text{def}}{=} \max_{1\le i \le n} L_i$ and $\mathbf{L}_{i;j}$ is the $j$th diagonal element of matrix $\mathbf{L}_i$. Parameters $\nu \in [1, n]$ and $\nu_1 \in [1, d]$ describe the level of heterogeneity over the nodes and coordinates respectively. If $\mathbf{L}_i$ matrices coincide, then $\nu = n$ and $\nu_1 = d$. On the other extreme, when the values of $\mathbf{L}_i$ are extremely non-uniform, we have $\nu \ll n$ and $\nu_1 \ll d$.

Notice that the quantity $\frac{\mathcal{L}_{\max}}{\mu n}$ in (6) and the quantity $\omega_{\max} + \frac{\mathcal{L}_{\max}}{\mu n}$ in (7) depend on compression operators $\mathcal{C}_i^k$ applied by the nodes. For the rest of the paper we are going to minimize these quantities with respect to the choice of $\mathcal{C}_i^k$ in such a way to minimize total communication complexity of the proposed distributed methods. We specialize compressors $\mathcal{C}_i$ to two different extensions of standard quantization and optimize with respect to compression parameters.

## 4 Block Quantization

We now present our first extension to standard quantization in order to properly capture the matrix smoothness information. Instead of having a single quantization parameter (e.g. number of levels) for all coordinates, here we divide the space $\mathbb{R}^d$ into $B \in \{1, 2, \ldots, d\}$ blocks as $\mathbb{R}^d = \mathbb{R}^{d_1} \times \mathbb{R}^{d_2} \times \cdots \times \mathbb{R}^{d_B}$ and for each subspace $\mathbb{R}^{d_l}$, $l \in [B]$ we apply standard quantization independently from

---

[2]this can be shown by specializing Theorem 1 or Theorem 2 of Safaryan et al. [2021] to scalar smoothness setup and interpolation regime, namely $\mathbf{L}_i = L_i \mathbf{I}$ and $\|\nabla f_i(x^*)\| = 0$ for all $i \in [n]$.

other blocks with different number of levels $s_l$. Thus, for any $l \in [B]$ we allocate one parameter $s_l$ for $l^{th}$ block of $x \in \mathbb{R}^d$. Hence quantization is applied block-wise: for each block we send the norm $\|x^l\|$ of the block $x^l \in \mathbb{R}^{d_l}$ and all entries within this block are quantized with levels $\{0, \frac{1}{s_l}, \frac{2}{s_l}, \ldots, 1\}$. In the special case of $B = 1$, we get the standard quantization of Alistarh et al. [2017].

To get rid of the constraints on $s_l$ to be integers, instead of working with the number of levels $s_l$, we introduce the size of the quantization step $h_l = \frac{1}{s_l}$ and allow them to take any positive values (even bigger than 1). Thus, for each block $l \in [B]$ we quantize with respect to levels $\{0, h_l, 2h_l, \ldots \}$.

**Definition 3** (Block Quantization). For a given number of blocks $B \in [d]$ and fixed quantization steps $h = (h_1, \ldots, h_B)$, define block-wise quantization operator $\mathcal{Q}_h^B \colon \mathbb{R}^d \to \mathbb{R}^d$ as follows:

$$\left[\mathcal{Q}_h^B(x)\right]_t \overset{\text{def}}{=} \|x^l\| \cdot \text{sign}(x_t) \cdot \xi_l\left(\frac{|x_t|}{\|x^l\|}\right),$$

where $t = (l-1)B + j$, $x \in \mathbb{R}^d$, $j \in [d_l]$, $l \in [B]$ and $\xi_l(v)$ for $v \geq 0$ is defined via the quantization levels $\{0, h_l, 2h_l, \ldots \}$ as follows: if $kh_l \leq v < (k+1)h_l$ for some $k \in \{0, 1, 2, \ldots \}$, then

$$\xi_l(v) \overset{\text{def}}{=} \begin{cases} kh_l & \text{with probability} \quad k+1-\frac{v}{h_l}, \\ (k+1)h_l & \text{with probability} \quad \frac{v}{h_l}-k. \end{cases} \tag{9}$$

Note that $\mathcal{Q}_h^B$ is an unbiased compression operator as $\mathbb{E}\left[\xi_j(v)\right] = v$ for any $v \geq 0$. To communicate a vector of the form $\mathcal{Q}_h^B(x)$, we encode each block $\left[\mathcal{Q}_h^B(x)\right]^l \in \mathbb{R}^{d_l}$ using Elias $\omega$-coding as in the standard quantization scheme [Alistarh et al., 2017]. Hence, for each block $l \in [B]$ we need to send $\widetilde{\mathcal{O}}(\frac{1}{h_l^2} + \frac{\sqrt{d_l}}{h_l})$ bits and one floating point number for $\|x^l\|$. Overall, the number of encoding bits for $\mathcal{Q}_h^B(x)$ (up to constant and log factors) can be given by $\sum_{l=1}^B (\frac{1}{h_l^2} + \frac{\sqrt{d_l}}{h_l}) + B$. As for the compression noise, we prove in the Appendix the following upper bound for $\mathcal{L}(\mathcal{Q}_h^B, \mathbf{L})$:

$$\mathcal{L}(\mathcal{Q}_h^B, \mathbf{L}) \leq \max_{1 \leq l \leq B} h_l \| \text{Diag}(\mathbf{L}^{ll})\|, \tag{10}$$

where $\mathbf{L}^{ll}$ is the $l^{th}$ diagonal block matrix of $\mathbf{L}$ with sizes $d_l \times d_l$. Next, we are going to minimize communication complexity of DCGD+ and DIANA+ by optimizing parameters of block quantization.

**4.1. DCGD+ with block quantization.** We fix the number of blocks $B \in [d]$ for all nodes $i \in [n]$ and allow each node to apply different block quantization operator $\mathcal{Q}_{h_i}^B$ with quantization steps $h_i = (h_{i,1}, \ldots, h_{i,B})$. To minimize communication complexity of DCGD+, we need to minimize $\mathcal{L}_{\max}$ subject to the communication constraint mentioned above. Since $\mathcal{L}_{\max} = \max_{i \in [n]} \mathcal{L}(\mathcal{C}_i, \mathbf{L}_i)$, each node $i \in [n]$ can minimize the impact of its own compression by minimizing $\mathcal{L}(\mathcal{C}_i, \mathbf{L}_i)$ based on local smoothness matrix $\mathbf{L}_i$. This leads to the following optimization problem for finding optimal values of $h_i$ for each node $i \in [n]$:

$$\min_{h \in \mathbb{R}^B} \max_{1 \leq l \leq B} h_l \| \text{Diag}(\mathbf{L}_i^{ll})\|, \quad s.t. \ \sum_{l=1}^B (\frac{1}{h_l^2} + \frac{\sqrt{d_l}}{h_l}) + B = \beta, \ h_l > 0, \ l \in [B], \tag{11}$$

where $\beta$ is the "budget" of communication: Larger $\beta$ leads to finer quantization levels. Note that the constraint in (11) depends monotonically from each $h_l$. Therefore, the optimum is attained when $h_l \| \text{Diag}(\mathbf{L}_i^{ll})\|$ is uniform over $l \in [B]$. Thus, the solution to this problem is given by $h_{i,l} = \frac{\delta_{i,B}}{\| \text{Diag}(\mathbf{L}_i^{ll})\|}$, where $\delta_{i,B} \geq 0$ is uniquely determined by the constraint equality of (11) as the only positive solution of $\delta_{i,B}^2 - \delta_{i,B} \frac{dT_{i,B}}{\beta - B} - \frac{dT_{i,1}^2}{\beta - B} = 0$, which implies $\delta_{i,B} = \frac{dT_{i,B}}{2(\beta - B)} + \sqrt{\frac{d^2 T_{i,B}^2}{4(\beta - B)^2} + \frac{dT_{i,1}^2}{\beta - B}} \leq \frac{d}{\beta - B} T_{i,B} + \sqrt{\frac{d}{\beta - B}} T_{i,1}$, where $T_{i,B} \overset{\text{def}}{=} \frac{1}{d} \sum_{l=1}^B \sqrt{d_l} \| \text{Diag}(\mathbf{L}_i^{ll})\|$. If this solution of quantization steps $h_i$ is used by all nodes $i \in [n]$, then we show reduction in communication complexity by a factor of $\mathcal{O}(n)$.

**Theorem 3.** *Assume $n = \mathcal{O}(\sqrt{d})$ and both $\nu, \nu_1$ are $\mathcal{O}(1)$. Then DCGD+ using block quantization with $B = n$ blocks, $d_l = \mathcal{O}(d/n)$ block sizes for all $l \in [n]$ and quantization steps $h_{i,l} = \delta_{i,B}/\| \text{Diag}(\mathbf{L}_i^{ll})\|$ with $\beta = \mathcal{O}(d/n)$ reduces overall communication complexity by a factor of $\mathcal{O}(n)$ compared to DCGD using $B = 1$ single block quantization. Formally, to guarantee $\varepsilon > 0$ accuracy, the communication complexity of DCGD+ is $\mathcal{O}\left(\frac{d}{n} \frac{L_{\max}}{\mu} \log \frac{1}{\varepsilon}\right)$, which is $\mathcal{O}(n)$ times smaller over DCGD.*

**4.2. DIANA+ with block quantization.** For the rate (7) of DIANA+, we need to optimize $\omega_{\max} + \frac{\mathcal{L}_{\max}}{n\mu}$ part of the complexity under the same communication constraint used in (11). Since

$$\max_{i\in[n]}\left(\omega_i + \frac{\mathcal{L}_i}{n\mu}\right) \leq \omega_{\max} + \frac{\mathcal{L}_{\max}}{n\mu} \leq 2\max_{i\in[n]}\left(\omega_i + \frac{\mathcal{L}_i}{n\mu}\right), \qquad (12)$$

we can decompose the problem into subproblems for each node $i$ to optimize $\omega_i + \frac{\mathcal{L}_i}{n\mu}$ with respect to its own quantization parameters $h_i$. Analogously, this leads to the following optimization problem for finding optimal values of $h_i$ for each node $i \in [n]$:

$$\min_{h\in\mathbb{R}^B}\max_{1\leq l\leq B} h_l\left(\sqrt{d_l} + \frac{1}{\mu n}\|\operatorname{\mathbf{Diag}}(\mathbf{L}_i^{ll})\|\right), \ s.t. \ \sum_{l=1}^{B}(\frac{1}{h_l^2} + \frac{\sqrt{d_l}}{h_l}) + B = \beta, \ h_l > 0, \quad (13)$$

which can be solved with a similar argument as done for (11). Details are deferred to the Appendix.

**Theorem 4.** *Assume $n = \mathcal{O}(\sqrt{d})$ and both $\nu, \nu_1$ are $\mathcal{O}(1)$. Then DIANA+ using block quantization with $B = n$ blocks, $d_l = \mathcal{O}(d/n)$ block sizes for all $l \in [n]$ and $h_{i,l}$ quantization steps (solution to (13)) with $\beta = \mathcal{O}(d/n)$ reduces overall communication complexity by a factor of $\mathcal{O}(n)$ compared to DIANA using $B = 1$ single block quantization. Formally, to guarantee $\varepsilon > 0$ accuracy, the communication complexity of DIANA+ is $\mathcal{O}\left(\left(nd + \sqrt{\frac{d}{n}}\frac{L_{\max}}{\mu}\right)\log\frac{1}{\varepsilon}\right)$, which (ignoring $n$ summand in the complexity) is $\mathcal{O}(n)$ times smaller over DIANA.*

# 5 Quantization with Varying Steps

Our second extension of standard quantization scheme is to allow different quantization steps for all coordinates $\{1, 2, \ldots, d\}$. In other words, for each coordinate $j \in [d]$ we quantize with respect to levels $\{0, h_j, 2h_j, \ldots\}$. The standard quantization [Alistarh et al., 2017] is the special case when $h_j = \frac{1}{s}$ for all $j \in [d]$, where $s$ is the number of quantization levels.

**Definition 4** (Quantization with varying steps). For fixed quantization steps $h = (h_1, \ldots, h_d)^\top \in \mathbb{R}^d$, define quantization operator $\mathcal{Q}_h : \mathbb{R}^d \to \mathbb{R}^d$ as follows:

$$[\mathcal{Q}_h(x)]_j = \|x\| \cdot \operatorname{sign}(x_j) \cdot \xi_j\left(\frac{|x_j|}{\|x\|}\right), \quad x \in \mathbb{R}^d, \ j = 1, 2, \ldots, d,$$

where $\xi_j$ is defined via the quantization levels $\{0, h_j, 2h_j, \ldots\}$ as in (9).

Note that compression operator $\mathcal{Q}_h$ is unbiased as $\mathbb{E}[\xi_j(v)] = v$ for any $v \geq 0$ and is not a special case of block quantization defined earlier. To understand how the number of encoding bits of $\mathcal{Q}_h(x)$ depends on $h$ exactly seems challenging, since it depends on the actual encoding scheme (i.e. binary representation of compressed information). Besides, even if we fix binary mapping, the closed form expression of total amount of bits is too complicated to be utilized in the further analysis. We provide theoretical arguments and clear numerical evidence that $\|h^{-1}\| = \sqrt{\sum_{j=1}^{d} h_j^{-2}}$ is a reasonable proxy for the number of encoding bits for compressor $\mathcal{Q}_h$.

**Assumption 3.** For any input vector $x \in \mathbb{R}^d$ and quantization steps $h \in \mathbb{R}^d$, compressed vector $\mathcal{Q}_h(x)$ can be encoded with $\mathcal{O}(\|h^{-1}\|)$ number of bits.

First, consider the special case when all quantization steps are the same, i.e. $h_j = \frac{1}{s}$. Then $\|h^{-1}\| = s\sqrt{d}$ recovers the dominant part (provided $s = \mathcal{O}(\sqrt{d})$) in $\widetilde{\mathcal{O}}(s^2 + s\sqrt{d})$ showing total amount of bits for standard quantization scheme. Second, in the Appendix we present an encoding scheme which (up to constant and $\log d$ factors) requires $\mathbb{E}[\psi(\|\hat{x}\|_0)] + \|h^{-1}\|$ number of bits in expectation to communicate $\hat{x} = \mathcal{Q}_h(x)$, where $\psi(\tau) \stackrel{\text{def}}{=} dH_2(\tau/d) + \tau \leq d\log 3$, if $\tau \in [0, d]$ and $H_2$ is the binary entropy function. Note that, based on the definition (9), increasing quantization steps $h_j$ forces more sparsity in $\hat{x}$ and hence reduces $\|\hat{x}\|_0$. Thus, $\|\hat{x}\|_0$ and hence $\psi(\|\hat{x}\|_0)$ (notice that $\psi(0) = 0$) are proportional to $\|h^{-1}\|$. Furthermore, we present a numerical experiment which shows that the number of encoding bits of $\mathcal{Q}_h(x)$ and $\|h^{-1}\|$ are positively correlated.

Hence, in the further analysis, we fix the number of encoding bits of $\mathcal{Q}_h(x)$ by the constraint $\|h^{-1}\| = \beta$ for some parameter $\beta > 0$. As for the variance induced by the compression operator $\mathcal{Q}_h$, we prove the following upper bound for $\mathcal{L}(\mathcal{Q}_h, \mathbf{L})$:

$$\mathcal{L}(\mathcal{Q}_h, \mathbf{L}) \leq \|\operatorname{\mathbf{Diag}}(\mathbf{L})h\|. \qquad (14)$$

**5.1. DCGD+ with varying quantization steps.** Now, we optimize the rate (6) of DCGD+ with respect to quantization steps $h_i = (h_{i;1}, h_{i;2}, \ldots, h_{i;d})$ of compressor $\mathcal{Q}_{h_i}$ controlled by $i^{th}$ node for all $i \in [n]$. The term in (6) affected by the compression is $\mathcal{L}_{\max} = \max_{i \in [n]} \mathcal{L}(\mathcal{C}_i, \mathbf{L}_i)$, which implies that each node $i \in [n]$ can minimize the impact of its own compression by minimizing $\mathcal{L}(\mathcal{C}_i, \mathbf{L}_i)$ based on local smoothness matrix $\mathbf{L}_i$. Based on the upper bound (14) and communication constraint given by $\|h^{-1}\| = \beta$ for some $\beta > 0$, we get the following optimization problem to choose the optimal quantization parameters $h_i$ for node $i \in [n]$:

$$\min_{h \in \mathbb{R}^d} \| \mathbf{Diag}(\mathbf{L}_i)h\|, \quad s.t. \|h^{-1}\| = \beta, \ h_j > 0, \ j \in [d]. \tag{15}$$

This problem has the following closed form solution due to KKT conditions (see Appendix):

$$h_{i;j} = \frac{1}{\beta} \sqrt{\frac{\sum_{t=1}^d \mathbf{L}_{i;t}}{\mathbf{L}_{i;j}}}, \quad i \in [n], \ j \in [d]. \tag{16}$$

With this choice of quantization steps we save $\mathcal{O}(\min(n, d))$ times in communication.

**Theorem 5.** *Assume both $\nu, \nu_1$ are $\mathcal{O}(1)$ and $\beta = \mathcal{O}(d/n)$. Then DCGD+ using quantization with varying steps (25) for all $i \in [n]$ reduces overall communication complexity by a factor of $\mathcal{O}(\min(n, d))$ compared to the baseline of DCGD. Formally, the iteration complexity (6) can be upper bounded as $\frac{L}{\mu} + \frac{\mathcal{L}_{\max}}{n\mu} \leq \frac{\nu}{n} \frac{L_{\max}}{\mu} + \frac{\nu_1}{\beta} \frac{L_{\max}}{n\mu} = \mathcal{O}\left(\frac{1}{n} \frac{L_{\max}}{\mu} + \frac{1}{d} \frac{L_{\max}}{\mu}\right)$, which is $\min(n, d)$ times smaller than the one for DCGD. As both methods communicate $\mathcal{O}(d/n)$ bits per node per iteration, we get $\min(n, d)$ times savings in communication complexity.*

**5.2. DIANA+ with varying quantization steps.** Based on (12), each node $i \in [n]$ optimizes $\omega_i + \frac{\mathcal{L}_i}{n\mu}$ with respect to its quantization parameters $h_i$, which is equivalent to the problem

$$\min_{h \in \mathbb{R}^d} \sum_{j=1}^d \left(1 + A_{ij}^2\right) h_j^2, \quad s.t. \|h^{-1}\| = \beta, \ h_j > 0, j \in [d] \tag{17}$$

where $A_{ij} \stackrel{\text{def}}{=} \mathbf{L}_{i;j}/n\mu$. Due to the KKT conditions (see Appendix), we get the following solution

$$h_{i;j} = \frac{1}{\beta} \sqrt{\frac{\sum_{t=1}^d \sqrt{1 + A_{it}^2}}{\sqrt{1 + A_{ij}^2}}}. \tag{18}$$

With this choice of quantization steps we save $\mathcal{O}(\min(n, d))$ times in communication.

**Theorem 6.** *Assume both $\nu, \nu_1$ are $\mathcal{O}(1)$ and $\beta = \mathcal{O}(d/n)$. Then DIANA+ using quantization with varying steps (18) for all $i \in [n]$ reduces overall communication complexity by a factor of $\mathcal{O}(\min(n, d))$ compared to the baseline of DIANA. Formally, the iteration complexity (7) can be upper bounded as $\omega_{\max} + \frac{L}{\mu} + \frac{\mathcal{L}_{\max}}{n\mu} \leq \frac{\sqrt{2}d}{\beta} + \frac{\nu}{n} \frac{L_{\max}}{\mu} + \frac{\sqrt{2}\nu_1}{\beta n} \frac{L_{\max}}{\mu} = \mathcal{O}\left(n + \frac{1}{n} \frac{L_{\max}}{\mu} + \frac{1}{d} \frac{L_{\max}}{\mu}\right)$, which is $\min(n, d)$ times smaller than the one for DIANA (ignoring negligible term $n$).*

## 6 Experiments

**6.1. Setup.** In this section we present two key experiments. Additional experiments can be found in the Appendix. We conduct a range of experiments with several datasets from the LibSVM repository [Chang and Lin, 2011] on the $\ell_2$-regularized logistic regression problem (1):

$$\min_{x \in \mathbb{R}^d} \frac{1}{n} \sum_{i=1}^n f_i(x), \quad f_i(x) = \frac{1}{m} \sum_{t=1}^m \log(1 + \exp(-b_{i,t} \mathbf{A}_{i,t}^\top x)) + \frac{\lambda}{2} \|x\|^2,$$

where $\mathbf{A}_{i,t}$ are data points sorted based on their norms before allocating to local workers for the heterogeneity. The experiments are performed on a workstation with Intel(R) Xeon(R) Gold 6246 CPU @ 3.30GHz cores. The `gather` and `broadcast` operations for the communications between master and workers are implemented based on the MPI4PY library [Dalcín et al., 2005] and each CPU core is treated as a local worker. For each dataset, we run each algorithm multiples times with 5 random seeds for each worker. Due to space limitations, we present only two of our experiments here deferring the remaining experiments along with experimental details in the Appendix.

**6.2. Comparison to standard quantization techniques.** In our first experiment, we compare smoothness-aware DCGD+ and DIANA+ methods with our varying-step quantization technique

(`quant+`) to the original DCGD [Khirirat et al., 2018] and DIANA [Mishchenko et al., 2019] methods with the standard quantization technique (`quant`) of Alistarh et al. [2017]. Figure 1 demonstrates that DCGD+/DIANA+ with `quant+` lead to significant improvement in both transmitted megabytes and wall-clock time. An ablation study to disentangle the contributions of exploiting the smoothness matrix and utilizing varying number of levels can be found in Appendix B.

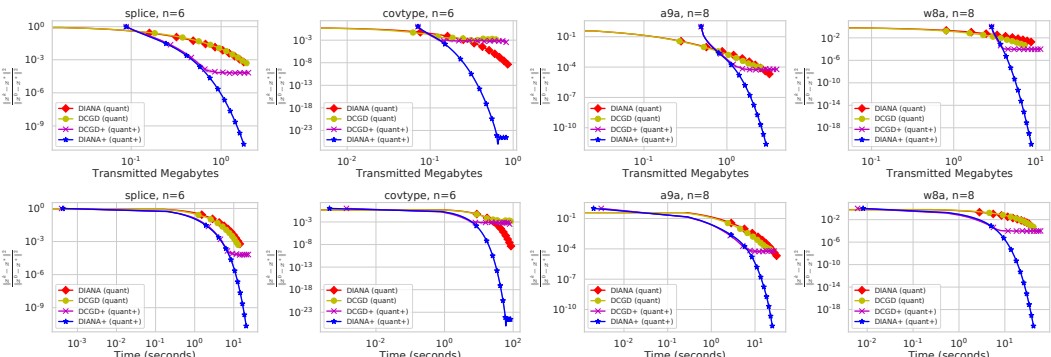

**Figure 1:** Comparison of smoothness-aware DCGD+/DIANA+ methods with varying-step quantization (`quant+`) to original DCGD/DIANA methods with standard quantization (`quant`). Note that in `quant+` workers need to send $\mathbf{L}_i^{1/2} \in \mathbb{R}^{d \times d}$ and quantization steps $h_i \in \mathbb{R}^d$ to the master before the training. This leads to extra costs in communication bits and time, which are taken into consideration.

**6.3. Comparison to matrix-smoothness-aware sparsification.** Second experiment is devoted to the performance of three smoothness-aware compression techniques —block quantization (`block quant+`) of Section 4, varying-step quantization (`quant+`) of Section 5 and smoothness-aware sparsification strategy (`rand-τ+`) of Safaryan et al. [2021]. All three compression techniques are shown to outperform the standard compression strategies by at most $\mathcal{O}(n)$ times in theory. For the sparsification, we use the optimal probabilities and the sampling size $\tau = d/n$ as suggested in Section 5.3 of [Safaryan et al., 2021]. The empirical results in Figure 2 illustrate that the varying-step quantization technique (`quant+`) is always better than the smoothness-aware sparsification [Safaryan et al., 2021], in terms of both communication cost and wall-clock time. Our block quantization technique also beats sparsification when the dimension of the model is relatively high.

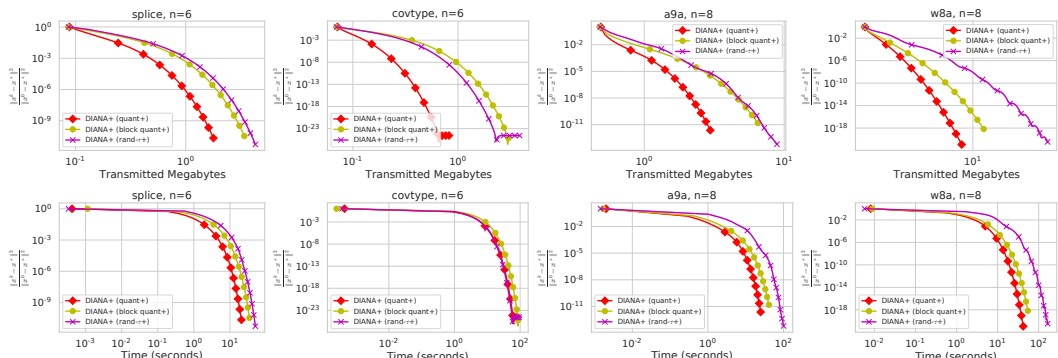

**Figure 2:** Comparison of three matrix-smoothness-aware compression techniques employed in DIANA+ method: varying-step quantization `quant+`, our variant of block quantization `block quant+`, and smoothness-aware sparsification `rand-τ+` of Safaryan et al. [2021].

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
