# Appendix

## A  Conclusions and Limitations

In this work we extended the matrix-smoothness-aware sparsification strategy of Safaryan et al. [2021] to arbitrary unbiased compression schemes. This significantly broadens the use of smoothness matrices in communication efficient distributed methods.

### A.1  Generalization and quantization

It is worth to mention that our results generalize those of Safaryan et al. [2021] in a tight manner. That is, we recover the same convergence guarantees as a special case. Indeed, if compression operators $\mathcal{C}_i$ are diagonal sketches $\mathbf{C}_i$ generated independently from others and via arbitrary samplings, then

$$
\begin{aligned}
\mathcal{L}_i &= \mathcal{L}(\mathbf{C}_i, \mathbf{L}_i) \\
&= \inf \left\{ \mathcal{L} \geq 0 \colon \mathbb{E} \left[ \|\mathbf{C}_i x - x\|_{\mathbf{L}_i}^2 \right] \leq \mathcal{L} \|x\|^2 \ \forall x \in \mathbb{R}^d \right\} \\
&= \inf \left\{ \mathcal{L} \geq 0 \colon x^\top \mathbb{E} \left[ (\mathbf{C}_i - \mathbf{I}) \mathbf{L}_i (\mathbf{C}_i - \mathbf{I}) \right] x \leq \mathcal{L} \|x\|^2 \ \forall x \in \mathbb{R}^d \right\} \\
&= \lambda_{\max} \left( \mathbb{E} \left[ (\mathbf{C}_i - \mathbf{I}) \mathbf{L}_i (\mathbf{C}_i - \mathbf{I}) \right] \right) \\
&= \lambda_{\max} \left( \mathbb{E} \left[ \mathbf{C}_i \mathbf{L}_i \mathbf{C}_i \right] - \mathbf{L}_i \right) \\
&= \lambda_{\max} (\overline{\mathbf{P}}_i \circ \mathbf{L}_i - \mathbf{L}_i) \\
&= \lambda_{\max} (\widetilde{\mathbf{P}}_i \circ \mathbf{L}_i),
\end{aligned}
$$

with the same probability matrices $\overline{\mathbf{P}}_i$ and $\widetilde{\mathbf{P}}_i$ defined in [Safaryan et al., 2021].

Further, we designed two novel quantization schemes (see Definitions 3 and 4) capable of properly utilizing matrix smoothness information of local loss functions in distributed optimization. We showed that the proposed quantization schemes can significantly outperform the key baselines both in theory and practice.

### A.2  Technical contributions

We make *two main technical contributions.*

First, we introduce the quantity $\mathcal{L}(\mathcal{C}, \mathbf{L})$ that properly captures *non-linear interaction* between the compressor $\mathcal{C}$ and the smoothness matrix $\mathbf{L}$. Due to the linearity of the sparsification (i.e., $\mathcal{C}(x) = \mathbf{C}x$), in previous work (Safaryan et al., 2021) it is easy to separate the sparsifier from the compressed gradient and combine it with the smoothness matrix:

$$
\|\mathbf{L}^{1/2} \mathcal{C}(\mathbf{L}^{\dagger 1/2} \nabla f(x))\|^2 = \nabla f(x)^\top \mathbf{L}^{\dagger 1/2} (\mathbf{C}\mathbf{L}\mathbf{C}) \mathbf{L}^{\dagger 1/2} \nabla f(x),
$$

where $\mathbf{C}\mathbf{L}\mathbf{C}$ shows a *linear interaction* between $\mathbf{C}$ and $\mathbf{L}$. Once we came up with the proper notion of $\mathcal{L}(\mathcal{C}, \mathbf{L})$ (we had other approaches before we found the "right" one), the proofs of Theorems 1 and 2 followed standard steps. Note that $\mathcal{L}(\mathcal{C}, \mathbf{L})$ *recovers the previous quantity* when the compressor is specialized to a sparsifier (see Sec A.1). This contribution may seem simple from hindsight, but it is not.

Our second technical contribution is the introduction of *two non-linear compressors* that provably benefit from smoothness matrices. Specifically, we formulate and analytically solve 4 intermediate optimization problems (11), (14), (16), (18) to find out the best parameter setting for each quantization scheme based on the smoothness information. In fact, sections 4 and 5 outline the technical difficulties we managed to overcome in order to get $\min(n, d)$ speedup factors in each case. *Our key contribution is the proposal of these two modified quantization schemes.* We adapt methods DCGD+ and DIANA+ to showcase the potential of our quantization strategies in reducing communication complexity. We chose DCGD as it is the simplest gradient type method with communication compression, and DIANA as it is the variance-reduced version of DCGD. Of course, one can apply our quantization techniques to other distributed methods and gain similar improvements (see Sec A.2). However, we are not attempting to (and can't) be exhaustive in this direction as there are many methods in the literature employing communication compression.

### A.3 Limitations and possible workarounds

Next, we discuss main limitations of our work.

- Note while in this paper we redesigned only two methods, DCGD+ and DIANA+, the modifications we suggest are not limited to these two methods and can be applied to other distributed methods. In particular, with a similar proof technique, ADIANA+ method of Safaryan et al. [2021] introduced with sparsification can also be extended to arbitrary unbiased compression operator using the new notion of $\mathcal{L}(\mathcal{C}, \mathbf{L})$.

- The computation or estimation of the smoothness matrix $\mathbf{L}_i$ requires addiotional prepro-cessing. For generalized linear models (GLM) (e.g., linear/logistic regression, SVM with smooth hinge loss) the matrix $\mathbf{L}_i$ can be *written in closed form using the local dataset* (see Lemma 1 of [Safaryan et al., 2021]). For example,

$$\mathbf{L}_i = \frac{1}{4m_i} \sum_{m=1}^{m_i} \mathbf{A}_{im}^\top \mathbf{A}_{im}$$

  for logistic regression, where $\{\mathbf{A}_{im} : m = 1, \ldots, m_i\}$ is the local data of device $i$. Beyond GLMs, $\mathbf{L}_i$ can be difficult to compute. Note that we do not claim that the proposed method would be practical for high-dimensional deep learning problems - but perhaps this will be overcome in future research. One possibility is to treat $\mathbf{L}_i$'s as hyper-parameters and learn some *rough approximations of the smoothness matrices from the first order information obtained by running a gradient type method*. This can be done initially as a preprocessing step, after which the matrices are considered "learned", and then our compression can be built and used.

- The server is required to store $d \times d$ matrices $\mathbf{L}_i^{1/2}$ for all nodes $i \in [n]$ and multiply them by sparse updates $\mathcal{C}_i^k(\mathbf{L}_i^{\dagger 1/2} \nabla f_i(x^k))$ in each iteration. Moreover, each node $i$ is required to store only its smoothness matrix $\mathbf{L}_i^{\dagger 1/2}$ and perform multiplication $\mathbf{L}_i^{\dagger 1/2} \nabla f_i(x^k)$ in each iterate. Hence, our methods are practical when either dimension $d$ is not too big or smoothness matrices $\mathbf{L}_i$ are of special structure (e.g., diagonal, low-rank).

- We did not analyze the compression of the smoothness matrix before communication as it is transferred **only once** before the training begins. Besides, we showed in our experiments that the overhead in communication cost is negligible when the number of iterations is large (the transmitted megabytes do not start from 0 in our plots).

  However, in practice, compressing the matrix $\mathbf{L}$ is a good idea. One option for that is to initially estimate a diagonal smoothness matrix that is as easy to communicate (still **only once**) as one full precision gradient. Another option is to directly apply compression to the matrix $\mathbf{L}$ so that the compressed matrix is an over-approximation. For example, let $\mathbf{L} = \sum_{k=1}^d \lambda_k u_k u_k^\top$ be the eigendecomposition of $\mathbf{L}$, where $\lambda_k$ is the $k^{th}$ largest eigenvalue corresponding to eigenvector $u_k$. Then

$$\mathbf{L} \preceq \sum_{k=1}^r \lambda_k u_k u_k^\top + \sum_{k=r+1}^d \lambda_{r+1} u_k u_k^\top = \sum_{k=1}^r (\lambda_k - \lambda_{r+1}) u_k u_k^\top + \lambda_{r+1} \mathbf{I}.$$

  The latter over-approximation (which serves as a smoothness matrix for $f$) can be transferred with $rd + 1$ floats where $r$ can be chosen small.

- For the sake of presentation, we analyzed both DCGD+ and DIANA+ when exact local gradients, $\nabla f_i$, can be computed by all nodes in each iteration. However, we believe that it is possible to extend the analysis to stochastic local gradient oracles. Current tools handling stochastic gradients can be easily applied to our matrix-smoothness-aware compression techniques.

- In our distributed methods we only compress uplink communication from nodes to the server, which is typically more bandwidth limited than downlink communication from the server to nodes. We believe that techniques that ensure compressed communication in both directions can be applied in our setting, too.

- We developed all our theory for strongly convex objectives. Extending the theory to convex and non-convex problems in a tight manner seems to be more challenging.

# B  Additional Experiments

In this section we provide additional experiments to highlight effectiveness of our approach.

## B.1  Setup

We run the experiments with several datasets listed in Table 2 from the LibSVM repository [Chang and Lin, 2011] on the $\ell_2$-regularized logistic regression problem described below:

$$\min_{x \in \mathbb{R}^d} \frac{1}{n} \sum_{i=1}^{n} f_i(x), \quad \text{where} \quad f_i(x) = \frac{1}{m} \sum_{t=1}^{m} \log(1 + \exp(-b_{i,t} \mathbf{A}_{i,t}^{\top} x)) + \frac{\lambda}{2} \|x\|^2,$$

where $x \in \mathbb{R}^d$, $\mathbf{A}_{i,l} \in \mathbb{R}^d$, $b_{i,l} \in \{-1, 1\}$ are the feature and label of $l$-th data point on the $i$-th worker, where the features of each $\mathbf{A}_{i,l}$ are rescaled into $[-1, 1]$. The data points are sorted based on their norms before allocating to local workers to ensure that the data split is heterogeneous. The experiments are performed on a workstation with Intel(R) Xeon(R) Gold 6246 CPU @ 3.30GHz cores. The `gather` and `broadcast` operations for the communications between master and workers are implemented based on the MPI4PY library [Dalcín et al., 2005] and each CPU core is treated as a local worker. We set $\lambda = 10^{-3}$ for all datasets. For each dataset, we run each algorithm multiples times with 5 random seeds for each worker.

**Table 2:** Information of the experiments on $\ell_2$-regularized logistic regression.

| Dataset | #Instances $N$ | Dimension $d$ | #Workers $n$ | #Instances/worker $m$ |
|---------|---------------|---------------|--------------|----------------------|
| german | 1,000 | 24 | 4 | 250 |
| svmguide3 | 1,243 | 21 | 4 | 310 |
| covtype | 581,012 | 54 | 6 | 145,253 |
| splice | 1,000 | 60 | 6 | 166 |
| w8a | 49,749 | 300 | 8 | 6,218 |
| a9a | 22,696 | 123 | 8 | 2,837 |

To implement Elias encoding and decoding, we utilize the EliasOmega library[3]. We compare the relative errors of different algorithms with respect to 3 measures: the number of iterations, the transmitted megabytes and wall-clock time. To be specific, the measured wall-clock time includes 1) the time of computation on each local worker in one iteration (e.g., local gradient computation, matrix multiplication, etc.); 2) the time of Elias coding and decoding; 3) the time of communication (`gather` and `broadcast`). It is worth noting that DCGD+ and DIANA+ require extra cost to transmit $\mathbf{L}_i^{1/2}$ beforehand. Moreover, when coupled with varying number of quantization levels, they also need to transmit $h_{i;j}$ before the start of training. These overheads are taken into consideration in our experimental results.

## B.2  Comparison to standard quantization techniques

First, we compare DCGD+/DIANA+ with the block quantization technique (`block quant+`) described in Section 4 to DCGD [Khirirat et al., 2018]/DIANA [Mishchenko et al., 2019] with the standard quantization technique (`quant`) in [Alistarh et al., 2017]. As shown in Figure 6, DCGD+ (`block quant+`) and DIANA+ (`block quant+`) outperform DCGD (`quant`) and DIANA (`quant`) when $d$ is larger. This is understandable because the extra cost on communication $B$ norms becomes neglectable when the dimension is relatively high given the number of blocks, where splitting the whole parameters into blocks makes more sense.

---

[3] `https://gist.github.com/robertofraile/483003`

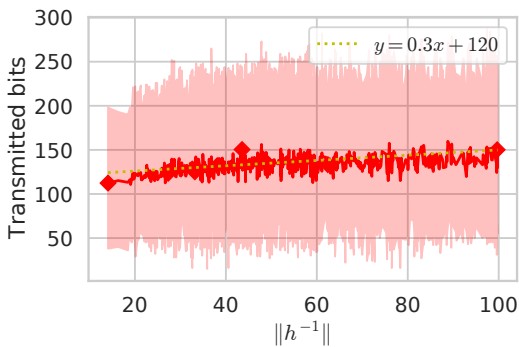

**Figure 3:** Experiment to verify the Assumption 3. We randomly generate 1000 quantization step vectors $h \in \mathbb{R}^{50}$, each component of $h$ is $h_j = |\tilde{h}_j|$ and $\tilde{h}_j$ is independently sampled from $\mathcal{N}(0, 1)$. For each $h$, we randomly generate multiple sparse vectors to quantize $x$, which is sampled from Poisson distribution with $\lambda = \{1, 10, 100\}$ and density $\{0.25, 0.5, 0.75, 1.0\}$.

Next, we compare DCGD+/DIANA+ with our second quantization technique (`quant+`) that has varying number of quantization steps per coordinate to DCGD (`quant`) and DIANA (`quant`). Figure 7 demonstrates that DCGD+ (`quant+`) and DIANA+ (`quant+`) lead to significant improvement.

### B.3 Ablation study of DIANA+ (`block quant+`) and DIANA+ (`quant+`)

As mentioned by Alistarh et al. [2017], combining DCGD and block quantization can improve its iteration complexity at the cost of transmitting extra $32B$ bits per iteration, which might also lead to better total communication complexity. Thus, the advantage of DIANA+ (`block quant+`) over DIANA (`quant`) may come from either splitting the features into blocks or exploiting the smoothness matrix. To further demistefy the improvement of DIANA+ (`block quant+`) , we compare the results of DIANA+ (`block quant+`), DIANA+ (`block quant`), DIANA (`block quant`) and DIANA (`quant`) in Figure 5. The difference between `block quant` and `block-quant+` is that the former one uses the same number of quantization levels for different blocks while the latter one uses varying numbers. It can be seen from Figure 5 that DIANA+ (`block-quant+`) consistently outperforms other methods because it optimally exploits the block structure and the smoothness matrix.

We also demonstrate that how DIANA+ perform with varying or fixed number of levels. As seen in Figure 6, the varying number of levels are beneficial on most of the datasets.

### B.4 Comparison to matrix-smoothness-aware sparsification

Moreover, we also compare the performance of three smoothness-aware compression techniques —block quantization (`block quant+`) of Section 4, varying-step quantization (`quant+`) of Section 5 and smoothness-aware sparsification strategy (`rand-`$\tau$`+`) of Safaryan et al. [2021]. All three compression techniques are shown to outperform the standard compression strategies by at most $\mathcal{O}(n)$ times in theory. For the sparsification, we use the optimal probabilities and the sampling size $\tau = d/n$ as suggested in Section 5.3 of [Safaryan et al., 2021]. The empirical results in Figure 8 illustrate that the varying-step quantization technique (`quant+`) is always better than the smoothness-aware sparsification [Safaryan et al., 2021], in terms of both communication cost and wall-clock time. Our block quantization technique also beats sparsification when the dimension of the model is relatively high.

### B.5 Numerical verification of Assumption 3

We provide a numerical experiment to verify Assumption 3 that $\|h^{-1}\|$ and the communicated bits are positively correlated. Figure 3 shows that the communicated bits and $\|h^{-1}\|$ are indeed positively correlated.

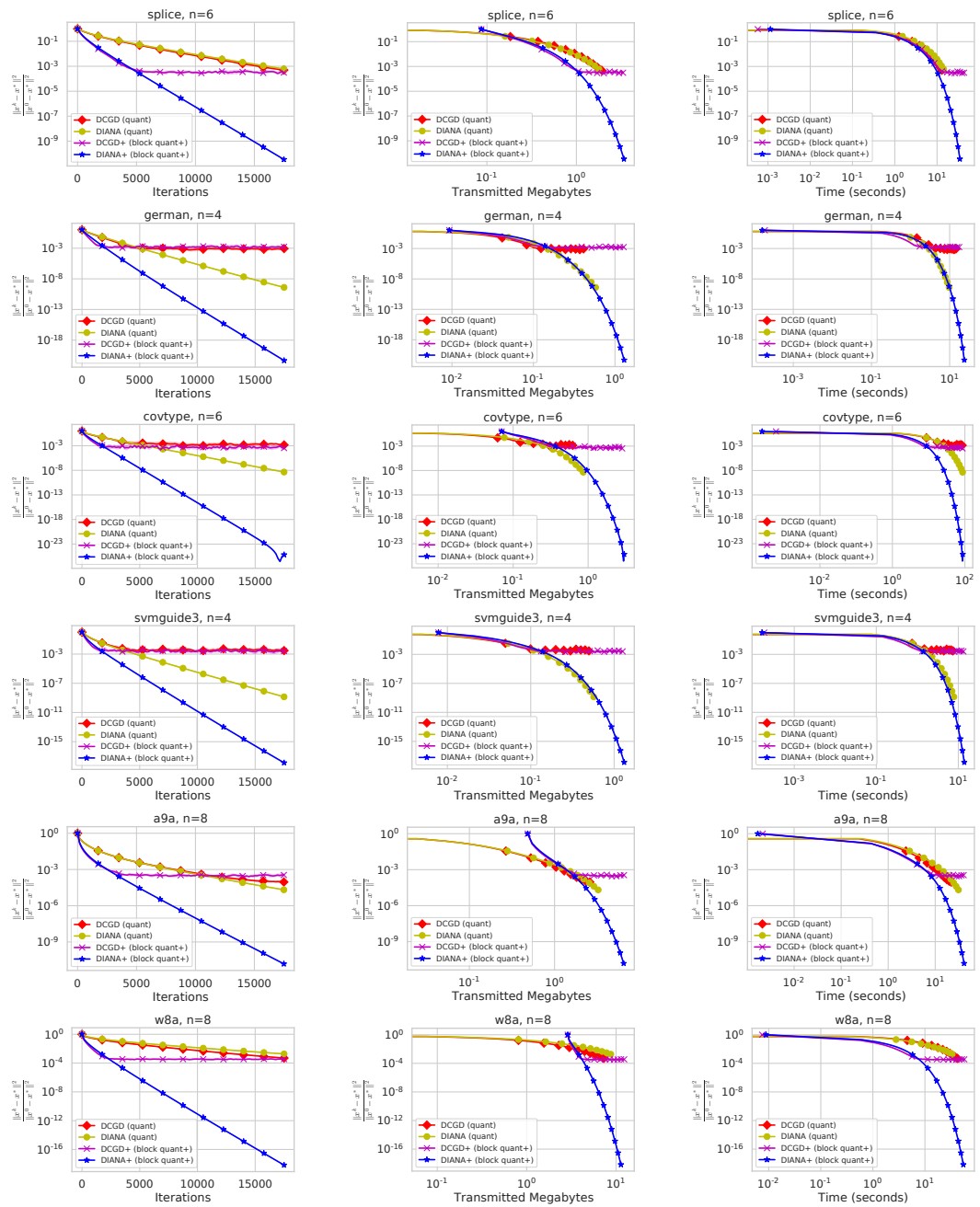

**Figure 4:** Comparison of DCGD+ (`block quant+`) and DIANA+ (`block quant+`) with DCGD (`quant`) and DIANA (`quant`).

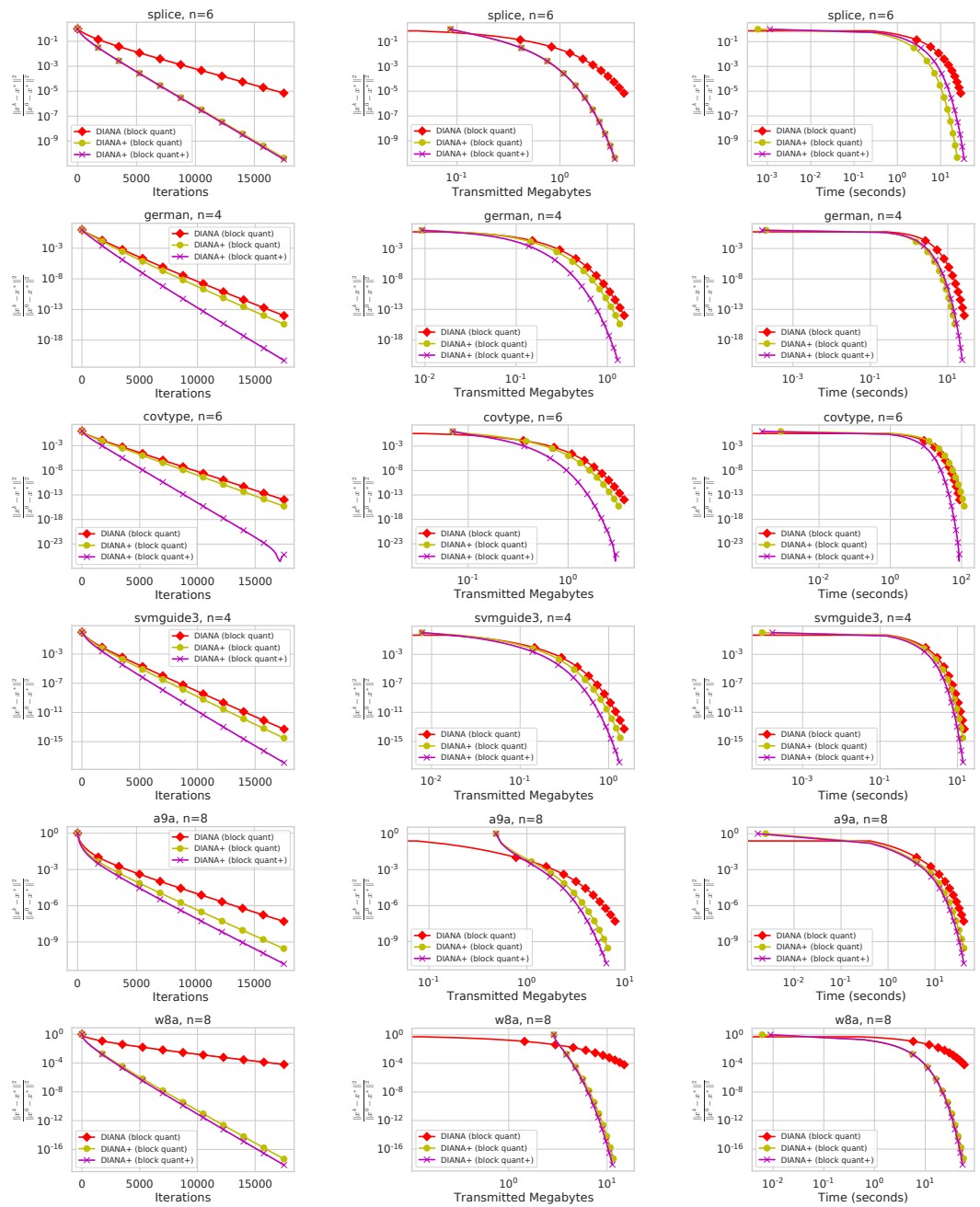

**Figure 5:** Comparison of DIANA+ (`block quant+`), DIANA+ (`block quant`), DIANA (`block quant`) and DIANA (`quant`).

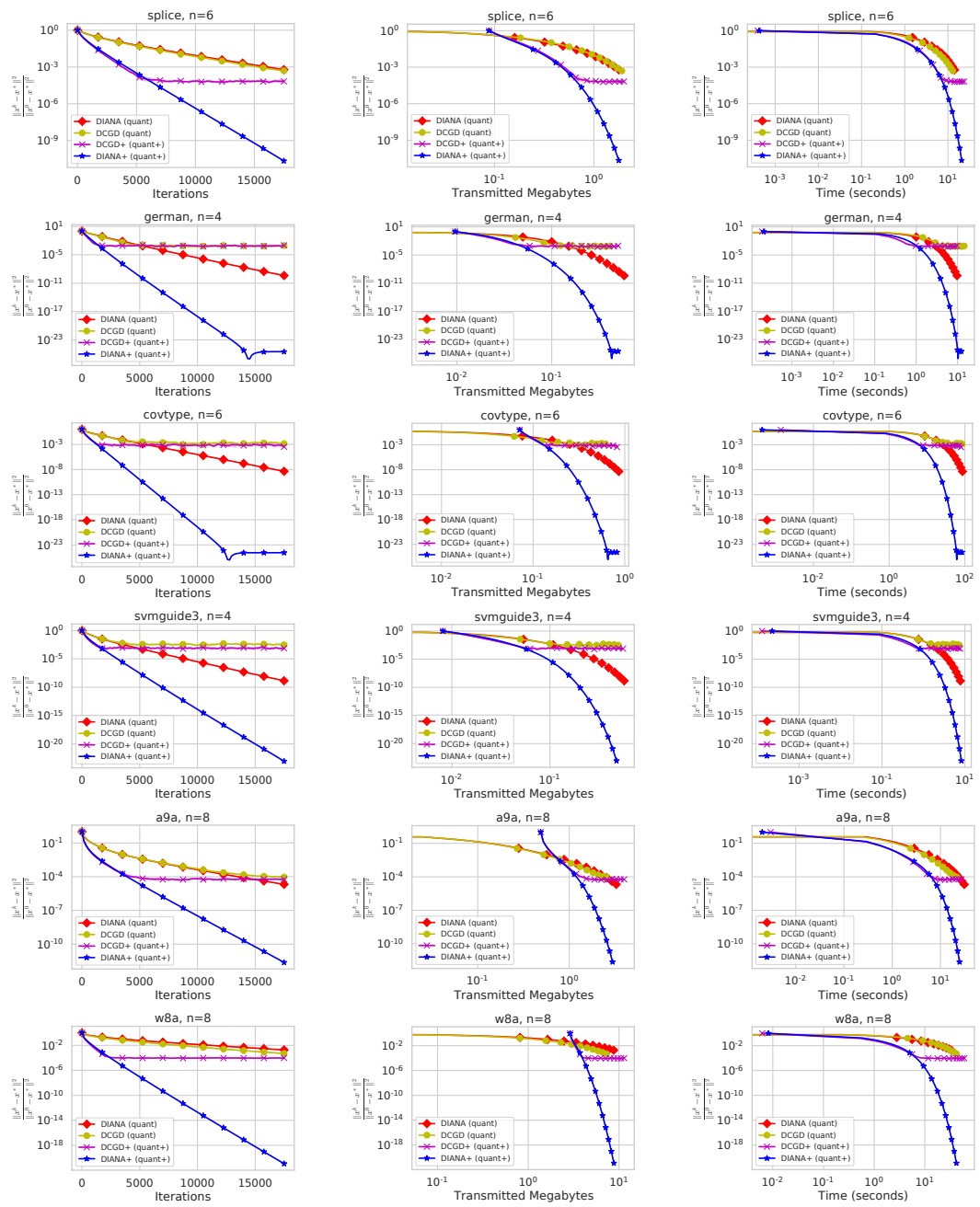

**Figure 6:** Comparison of DCGD+ (quant+) and DIANA+ (quant+) with DCGD (quant) and DIANA (quant).

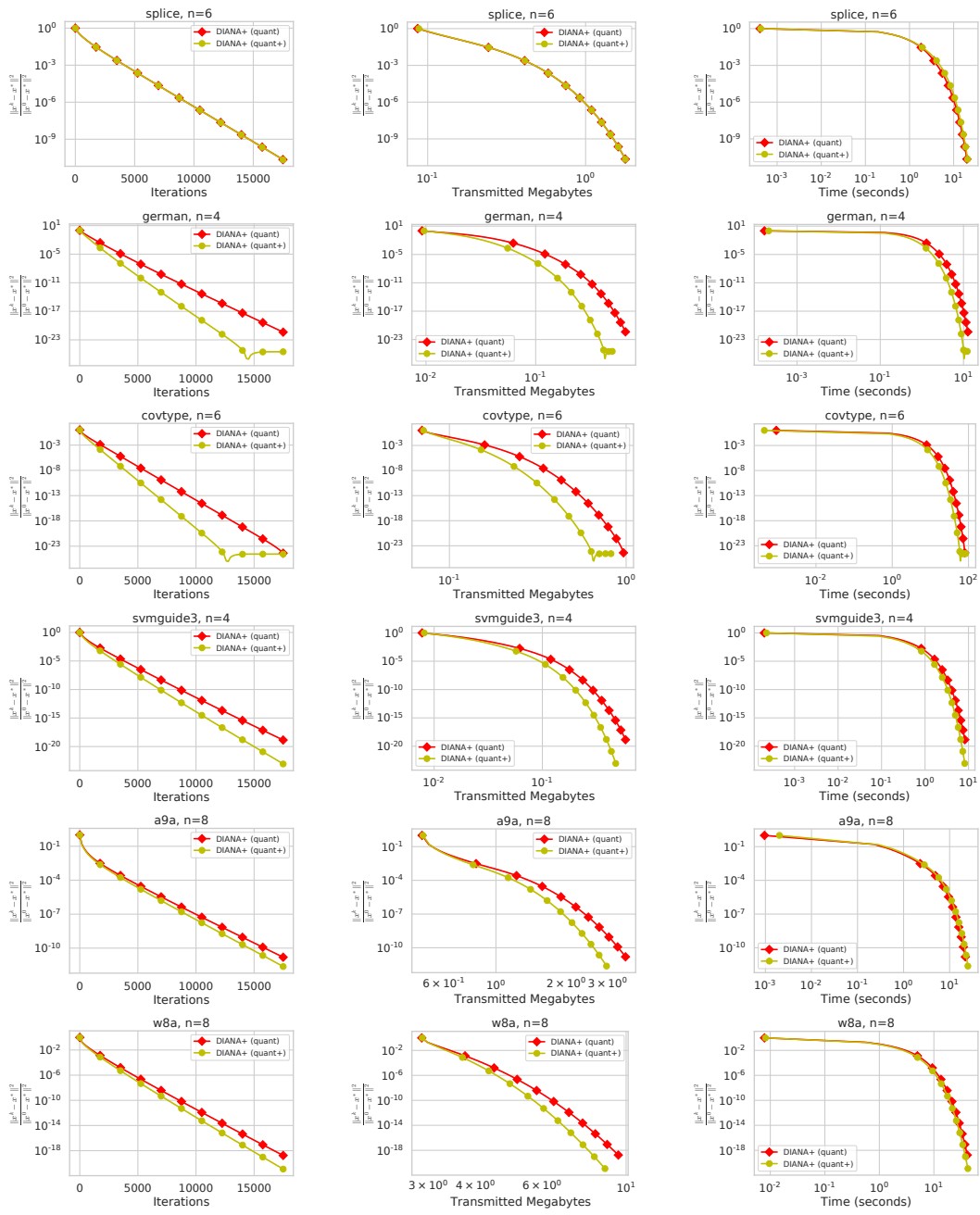

**Figure 7:** Comparison of DIANA+ with quantization that has varying or fixed number of levels.

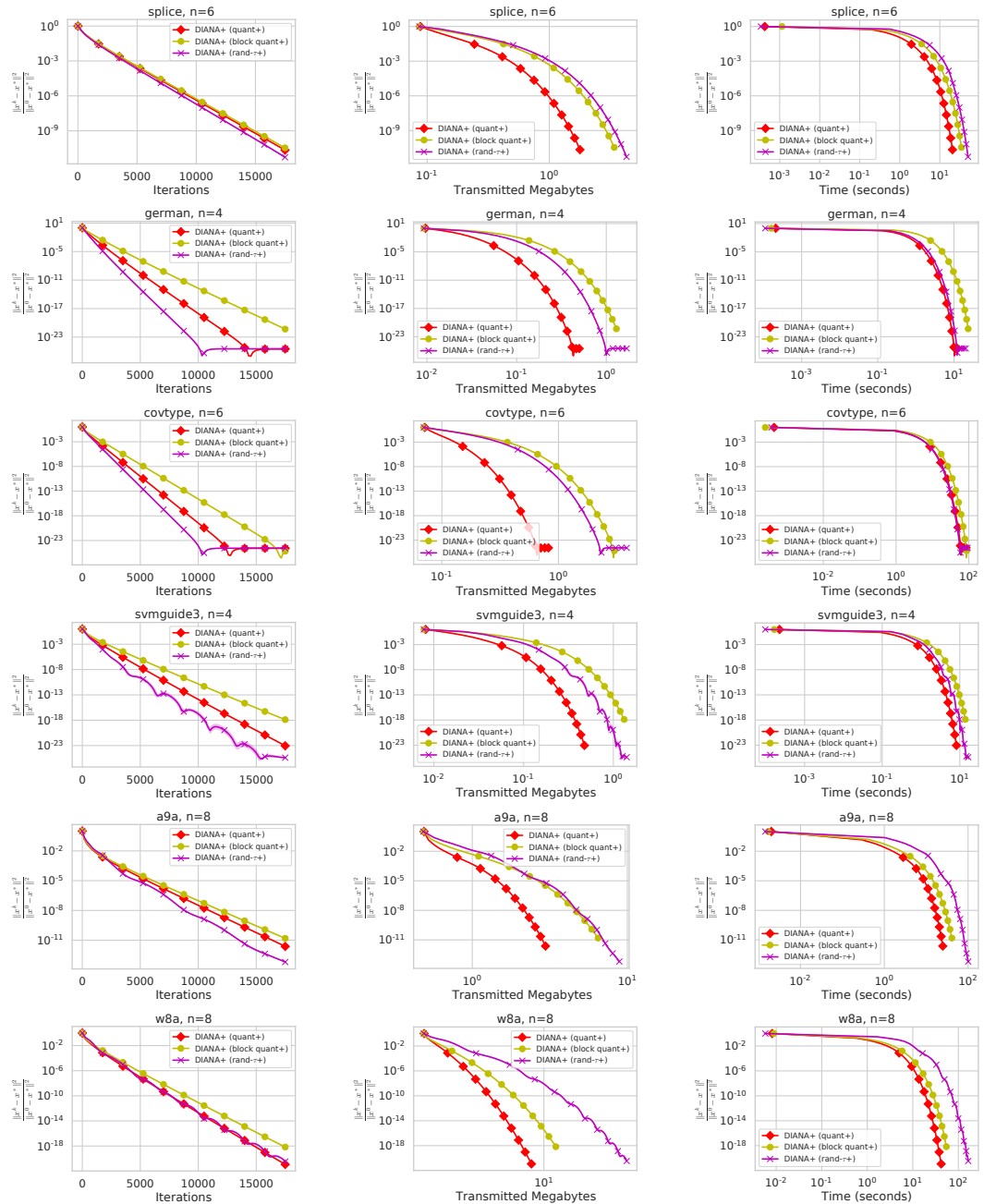

**Figure 8:** Comparison of smoothness-aware DCGD+/DIANA+ methods with varying-step quantization (`quant+`) to original DCGD/DIANA methods with standard quantization (`quant`). Note that in `quant+` workers need to send $\mathbf{L}_i^{1/2} \in \mathbb{R}^{d \times d}$ and quantization steps $h_i \in \mathbb{R}^d$ to the master before the training. This leads to extra costs in communication bits and time, which are taken into consideration.

## C Proofs for Section 3: Smoothness-Aware Distributed Methods with General Compressors

Here we provide the proofs of Theorem 1 and Theorem 2. Both proofs follow similar steps done for sparsification in [Safaryan et al., 2021].

### C.1 Proof of Theorem 1: DCGD+ with arbitrary unbiased compression

To simplify the notation, let us skip the iteration count $k$ in the derivations. We are going to estimate the quantity $\mathbb{E}\left[\|g(x) - \nabla f(x^*)\|^2\right]$ and establish the following bound for the gradient estimator $g(x) = \frac{1}{n}\sum_{i=1}^n \mathbf{L}_i^{1/2}\mathcal{C}_i(\mathbf{L}_i^{\dagger 1/2}\nabla f_i(x))$:

$$\mathbb{E}\left[\|g(x) - \nabla f(x^*)\|^2\right] \leq 2\left(L + \frac{2\mathcal{L}_{\max}}{n}\right)D_f(x, x^*) + \frac{2\sigma_+^*}{n},$$

where $D_f(x, x^*)$ is the Bregman divergence between $x$ and $x^*$ with respect to $f$. Due to Lemma E.3 [Hanzely and Richtárik, 2019], we have $\nabla f_i(x) = \mathbf{L}_i^{1/2}r_i$ for some $r_i$. Therefore,

$$\mathbb{E}\left[\mathbf{L}_i^{1/2}\mathcal{C}_i(\mathbf{L}_i^{\dagger 1/2}\mathbf{L}_i^{1/2}r_i)\right] = \mathbf{L}_i^{1/2}\mathbb{E}\left[\mathcal{C}_i(\mathbf{L}_i^{\dagger 1/2}\mathbf{L}_i^{1/2}r_i)\right] = \mathbf{L}_i^{1/2}\mathbf{L}_i^{\dagger 1/2}\mathbf{L}_i^{1/2}r_i = \mathbf{L}_i^{1/2}r_i = \nabla f_i(x), \quad (19)$$

which implies unbiasedness of the estimator $g(x)$, namely $\mathbb{E}[g(x)] = \nabla f(x)$. Note that:

$$
\begin{aligned}
\mathbb{E}\left[\|g(x) - \nabla f(x)\|^2\right] &= \mathbb{E}\left[\left\|\frac{1}{n}\sum_{i=1}^n \mathbf{L}_i^{1/2}\mathcal{C}_i(\mathbf{L}_i^{\dagger 1/2}\nabla f_i(x)) - \nabla f_i(x)\right\|^2\right] \\
&\overset{\clubsuit}{=} \frac{1}{n^2}\sum_{i=1}^n \mathbb{E}\left[\left\|\mathbf{L}_i^{1/2}\mathcal{C}_i(\mathbf{L}_i^{\dagger 1/2}\nabla f_i(x)) - \mathbf{L}_i^{1/2}\mathbf{L}_i^{\dagger 1/2}\nabla f_i(x)\right\|^2\right] \\
&= \frac{1}{n^2}\sum_{i=1}^n \mathbb{E}\left[\left\|\mathcal{C}_i(\mathbf{L}_i^{\dagger 1/2}\nabla f_i(x)) - \mathbf{L}_i^{\dagger 1/2}\nabla f_i(x)\right\|_{\mathbf{L}_i}^2\right] \\
&\overset{\spadesuit}{\leq} \frac{1}{n^2}\sum_{i=1}^n \mathcal{L}(\mathcal{C}_i, \mathbf{L}_i)\|\nabla f_i(x)\|_{\mathbf{L}_i^\dagger}^2 \\
&\overset{\diamond}{\leq} \frac{2}{n^2}\sum_{i=1}^n \mathcal{L}_i\|\nabla f_i(x) - \nabla f_i(x^*)\|_{\mathbf{L}_i^\dagger}^2 + \frac{2}{n^2}\sum_{i=1}^n \mathcal{L}_i\|\nabla f_i(x^*)\|_{\mathbf{L}_i^\dagger}^2 \\
&\overset{\star}{\leq} \frac{4}{n^2}\sum_{i=1}^n \mathcal{L}_i D_{f_i}(x, x^*) + \frac{2\sigma_+^*}{n} \\
&\leq \frac{4\mathcal{L}_{\max}}{n}D_f(x, x^*) + \frac{2\sigma_+^*}{n},
\end{aligned}
$$

where $\clubsuit$ is due to $\mathbb{E}\left[\mathbf{L}_i^{1/2}\mathcal{C}_i(\mathbf{L}_i^{\dagger 1/2}\nabla f_i(x))\right] = \nabla f_i(x)$ and $\nabla f_i(x) = \mathbf{L}_i^{1/2}\mathbf{L}_i^{\dagger 1/2}\mathbf{L}_i^{1/2}r_i = \mathbf{L}_i^{1/2}\mathbf{L}_i^{\dagger 1/2}\nabla f_i(x)$ based on (19) and $\nabla f_i(x) = \mathbf{L}_i^{1/2}r_i$ for some $r_i$. $\spadesuit$ can be directly obtained by noticing the definition of $\mathcal{L}(\mathcal{C}, \mathbf{L})$ in Table 3. $\diamond$ is based on the definition of $\mathcal{L}_i$ and the fact $\|x + y\|^2 \leq 2\|x\|^2 + 2\|y\|^2$. $\star$ is due to Lemma E.3 of Hanzely and Richtárik [2019] and the definition of $\sigma_+^*$ defined in Theorem 1. The inequality above together with convexity and $L$-smoothness of $f$ implies

$$
\begin{aligned}
\mathbb{E}\left[\|g(x) - \nabla f(x^*)\|^2\right] &= \|\nabla f(x) - \nabla f(x^*)\|^2 + \mathbb{E}\left[\|g(x) - \nabla f(x)\|^2\right] \\
&\leq 2LD_f(x, x^*) + \frac{4\mathcal{L}_{\max}}{n}D_f(x, x^*) + \frac{2\sigma_+^*}{n} \\
&\leq 2\left(L + \frac{2\mathcal{L}_{\max}}{n}\right)D_f(x, x^*) + \frac{2\sigma_+^*}{n}.
\end{aligned}
$$

Applying the result of Gorbunov et al. [2020] we conclude the proof.

## C.2 Proof of Theorem 2: DIANA+ with arbitrary unbiased compression

We start with the unbiasedness of the estimator

$$g^k = \frac{1}{n}\sum_{i=1}^{n} \mathbf{L}_i^{1/2}\mathcal{C}_i\left(\mathbf{L}_i^{\dagger 1/2}(\nabla f_i(x) - u_i^k)\right) + u_i^k.$$

In (19), we showed unbiasedness using inclusion $\nabla f_i(x^k) \in \mathrm{range}(\mathbf{L}_i)$. Assuming $u_i^k \in \mathrm{range}(\mathbf{L}_i)$ for all $k \geq 0$, we get $\nabla f_i(x^k) - u_i^k \in \mathrm{range}(\mathbf{L}_i)$ for all $k \geq 0$. Hence, in the same way we can show unbiasedness of $g^k$ as

$$
\begin{aligned}
\mathbb{E}_k\left[g^k\right] &= \frac{1}{n}\sum_{i=1}^{n} \mathbf{L}_i^{1/2}\mathbb{E}_k\left[\mathcal{C}_i\left(\mathbf{L}_i^{\dagger 1/2}(\nabla f_i(x) - u_i^k)\right)\right] + u_i^k \\
&= \frac{1}{n}\sum_{i=1}^{n} \mathbf{L}_i^{1/2}\mathbf{L}_i^{\dagger 1/2}(\nabla f_i(x) - u_i^k) + u_i^k \\
&= \frac{1}{n}\sum_{i=1}^{n} \nabla f_i(x^k) = \nabla f(x^k).
\end{aligned}
$$

The inclusion $u_i^k \in \mathrm{range}(\mathbf{L}_i)$ directly follows from the initialization $u_i^0 \in \mathrm{range}(\mathbf{L}_i)$ (see line 1 of Algorithm 2) and linear update rule of $u_i^{k+1} = u_i^k + \alpha \mathbf{L}_i^{1/2}\Delta_i^k$ (see line 5 of Algorithm 2). As both $\nabla f_i(x^k)$ and $u_i^k$ belong to $\mathrm{range}(\mathbf{L}_i)$, denote $\nabla f_i(x^k) - u_i^k = \mathbf{L}_i^{1/2}r_i^k$. Next we bound

$$
\begin{aligned}
\mathbb{E}\left[\|g(x) - \nabla f(x)\|^2\right] &= \mathbb{E}\left[\left\|\frac{1}{n}\sum_{i=1}^{n}\mathbf{L}_i^{1/2}\mathcal{C}_i\left(\mathbf{L}_i^{\dagger 1/2}(\nabla f_i(x) - u_i^k)\right) + u_i^k - \nabla f_i(x)\right\|^2\right] \\
&= \frac{1}{n^2}\sum_{i=1}^{n}\mathbb{E}\left[\left\|\mathbf{L}_i^{1/2}\mathcal{C}_i^k\left(\mathbf{L}_i^{\dagger 1/2}(\nabla f_i(x) - u_i^k)\right) - \mathbf{L}_i^{1/2}\mathbf{L}_i^{\dagger 1/2}(\nabla f_i(x) - u_i^k)\right\|^2\right] \\
&= \frac{1}{n^2}\sum_{i=1}^{n}\mathbb{E}\left[\left\|\mathcal{C}_i^k\left(\mathbf{L}_i^{\dagger 1/2}(\nabla f_i(x) - u_i^k)\right) - \mathbf{L}_i^{\dagger 1/2}(\nabla f_i(x) - u_i^k)\right\|_{\mathbf{L}_i}^2\right] \\
&\leq \frac{1}{n^2}\sum_{i=1}^{n}\mathcal{L}(\mathcal{C}_i,\mathbf{L}_i)\|\nabla f_i(x) - u_i^k\|_{\mathbf{L}_i^\dagger}^2 \\
&\leq \frac{2\mathcal{L}_{\max}}{n^2}\sum_{i=1}^{n}\|\nabla f_i(x) - \nabla f_i(x^*)\|_{\mathbf{L}_i^\dagger}^2 + \frac{2\mathcal{L}_{\max}}{n^2}\sum_{i=1}^{n}\|u_i^k - \nabla f_i(x^*)\|_{\mathbf{L}_i^\dagger}^2 \\
&\leq \frac{4\mathcal{L}_{\max}}{n^2}\sum_{i=1}^{n}D_{f_i}(x,x^*) + \frac{2\mathcal{L}_{\max}}{n}\sigma_+^k \\
&= \frac{4\mathcal{L}_{\max}}{n}D_f(x,x^*) + \frac{2\mathcal{L}_{\max}}{n}\sigma_+^k,
\end{aligned}
$$

where $\sigma_+^k \stackrel{\mathrm{def}}{=} \frac{1}{n}\sum_{i=1}^{n}\|u_i^k - \nabla f_i(x^*)\|_{\mathbf{L}_i^\dagger}^2$ is the error in the gradient learning process. To proceed, we need to establish contractive recurrence relation for $\sigma_+^k$. For each summand, we have

$$\mathbb{E}_k\left[\left\|u_i^{k+1}-\nabla f_i(x^*)\right\|_{\mathbf{L}_i^\dagger}^2\right]$$

$$=\ \mathbb{E}_k\left[\left\|u_i^k-\nabla f_i(x^*)+\alpha\overline{\Delta}_i^k\right\|_{\mathbf{L}_i^\dagger}^2\right]$$

$$=\ \left\|u_i^k-\nabla f_i(x^*)\right\|_{\mathbf{L}_i^\dagger}^2+2\alpha\left\langle u_i^k-\nabla f_i(x^*),\nabla f_i(x^k)-u_i^k\right\rangle_{\mathbf{L}_i^\dagger}+\alpha^2\mathbb{E}\left[\left\|\mathbf{L}_i^{1/2}\mathcal{C}_i\left(\mathbf{L}_i^{\dagger 1/2}(\nabla f_i(x)-u_i^k)\right)\right\|_{\mathbf{L}_i^\dagger}^2\right]$$

$$\leq\ \left\|u_i^k-\nabla f_i(x^*)\right\|_{\mathbf{L}_i^\dagger}^2+2\alpha\left\langle u_i^k-\nabla f_i(x^*),\nabla f_i(x^k)-u_i^k\right\rangle_{\mathbf{L}_i^\dagger}+\alpha^2\mathbb{E}\left[\left\|\mathcal{C}_i\left(\mathbf{L}_i^{\dagger 1/2}(\nabla f_i(x)-u_i^k)\right)\right\|^2\right]$$

$$\leq\ \left\|u_i^k-\nabla f_i(x^*)\right\|_{\mathbf{L}_i^\dagger}^2+2\alpha\left\langle u_i^k-\nabla f_i(x^*),\nabla f_i(x^k)-u_i^k\right\rangle_{\mathbf{L}_i^\dagger}+\alpha^2(1+\omega_i)\left\|\nabla f_i(x^k)-u_i^k\right\|_{\mathbf{L}_i^\dagger}^2$$

$$\leq\ \left\|u_i^k-\nabla f_i(x^*)\right\|_{\mathbf{L}_i^\dagger}^2+2\alpha\left\langle u_i^k-\nabla f_i(x^*),\nabla f_i(x^k)-u_i^k\right\rangle_{\mathbf{L}_i^\dagger}+\alpha\left\|\nabla f_i(x^k)-u_i^k\right\|_{\mathbf{L}_i^\dagger}^2$$

$$=\ (1-\alpha)\left\|u_i^k-\nabla f_i(x^*)\right\|_{\mathbf{L}_i^\dagger}^2+\alpha\left\|\nabla f_i(x^k)-\nabla f_i(x^*)\right\|_{\mathbf{L}_i^\dagger}^2,$$

$$\leq\ (1-\alpha)\left\|u_i^k-\nabla f_i(x^*)\right\|_{\mathbf{L}_i^\dagger}^2+2\alpha D_{f_i}(x^k,x^*),$$

where we used bounds $\alpha\leq\frac{1}{1+\omega_i}$ and $0\preceq\mathbf{L}_i^{1/2}\mathbf{L}_i^\dagger\mathbf{L}_i^{1/2}\preceq\mathbf{I}$. Thus, with $\alpha\leq\frac{1}{1+\omega_{\max}}$, the estimator $g^k$ of DIANA+ satisfies

$$\mathbb{E}_k\left[g^k\right]=\nabla f(x^k)$$

$$\mathbb{E}_k\left[\|g^k-\nabla f(x^*)\|^2\right]\leq 2\left(L+\frac{2\mathcal{L}_{\max}}{n}\right)D_f(x^k,x^*)+\frac{2\mathcal{L}_{\max}}{n}\sigma_+^k$$

$$\mathbb{E}_k\left[\sigma_+^{k+1}\right]\leq(1-\alpha)\sigma_+^k+2\alpha D_f(x^k,x^*).$$

Again, we apply the generic result of Gorbunov et al. [2020] to complete the proof.

# D    Proofs for Section 4: Block Quantization

Here we provide the missing proofs of Section 4.

## D.1    Proof of the variance bound (10)

Using Definition 3 of compression operator $\mathcal{Q}_h^B$, we have

$$
\begin{aligned}
\mathbb{E}\left[\|\mathcal{Q}_h^B(x) - x\|_{\mathbf{L}}^2\right] &= \sum_{l=1}^B \|x^l\|^2 \mathbb{E}\left[\left\|\xi_l\left(\frac{|x^l|}{\|x^l\|}\right) - \frac{|x^l|}{\|x^l\|}\right\|_{\mathbf{L}^{ll}}^2\right] \\
&\leq \sum_{l=1}^B \|x^l\|^2 \min\left(h_l^2 \sum_{j=1}^{d_l} \mathbf{L}_{jj}^{ll}, h_l\sqrt{\sum_{j=1}^{d_l}\left[\mathbf{L}_{jj}^{ll}\right]^2}\right) \\
&\leq \max_{1\leq l\leq B} \min\left(h_l^2 \sum_{j=1}^{d_l} \mathbf{L}_{jj}^{ll}, h_l\sqrt{\sum_{j=1}^{d_l}\left[\mathbf{L}_{jj}^{ll}\right]^2}\right) \|x\|^2 \\
&= \max_{1\leq l\leq B} \min\left(h_l^2 \|\mathbf{Diag}(\mathbf{L}^{ll})\|_1, h_l\|\mathbf{Diag}(\mathbf{L}^{ll})\|\right) \|x\|^2.
\end{aligned}
$$

From the definition of $\mathcal{L}(\mathcal{Q}_h^B, \mathbf{L})$ we get

$$
\mathcal{L}(\mathcal{Q}_h^B, \mathbf{L}) \leq \max_{1\leq l\leq B} \min\left(h_l^2\|\mathbf{Diag}(\mathbf{L}^{ll})\|_1, h_l\|\mathbf{Diag}(\mathbf{L}^{ll})\|\right),
$$

which implies (10) if we ignore the first term.

## D.2    Proof of Theorem 3: DCGD+ with block quantization

First, recall that quantization steps $h_i$ are given by

$$
h_{i,l} = \frac{\delta_{i,B}}{\|\mathbf{Diag}(\mathbf{L}_i^{ll})\|}, \; l \in [B], \quad \text{where } \delta_{i,B} \leq \frac{d}{\beta - B}T_{i,B} + \sqrt{\frac{d}{\beta - B}}T_{i,1}.
$$

Then, we have

$$
\begin{aligned}
\frac{\mathcal{L}_{\max}}{n} &= \frac{1}{n}\max_{i\in[n]}\mathcal{L}(\mathcal{Q}_{h_i}^B, \mathbf{L}_i) \\
&\leq \frac{1}{n}\max_{i\in[n]}\delta_{i,B} \\
&\leq \frac{1}{n}\max_{i\in[n]}\left[\frac{d}{\beta - B}T_{i,B} + \sqrt{\frac{d}{\beta - B}}T_{i,1}\right] \\
&\leq \left[\frac{d/n}{\beta - B}\right]\max_{i\in[n]}T_{i,B} + \sqrt{\frac{d/n}{\beta - B}}\max_{i\in[n]}\frac{T_{i,1}}{\sqrt{n}}.
\end{aligned}
$$

Set $\beta = d/n + n$ and $B = n$. Since $n = \mathcal{O}(\sqrt{d})$, we have $\beta = \mathcal{O}(d/n)$ and hence $\frac{d/n}{\beta - B} = 1$. For the sake of simplicity, assume $d_l = d/n$. Next

$$
\begin{aligned}
\frac{T_{i,1}}{\sqrt{n}} &\leq \frac{1}{\sqrt{nd}}\sum_{j=1}^d \mathbf{L}_{i;jj} \leq \frac{\nu_1 L_{\max}}{\sqrt{nd}} \\
T_{i,n} &\leq \frac{1}{d}\sum_{l=1}^n \sqrt{d_l}\sum_{j=1}^{d_l}\mathbf{L}_{jj}^{ll} \\
&= \frac{\max_{l\in[n]}\sqrt{d_l}}{d}\sum_{j=1}^d \mathbf{L}_{jj} \\
&= \frac{\max_{l\in[n]}\sqrt{d_l}}{d}\nu_1 L_{\max} \leq \frac{\nu_1 L_{\max}}{\sqrt{nd}}.
\end{aligned}
$$

Regardless of the choice $h_i$, using the following inequalities with respect to matrix order

$$\mathbf{L} \preceq \frac{1}{n} \sum_{i=1}^{n} \mathbf{L}_i, \quad \mathbf{L}_i \preceq n\mathbf{L}, \tag{20}$$

we bound $L$ as follows

$$L = \lambda_{\max}(\mathbf{L}) \overset{(20)}{\leq} \lambda_{\max}\left(\frac{1}{n}\sum_{i=1}^{n}\mathbf{L}_i\right) \leq \frac{1}{n}\sum_{i=1}^{n}\lambda_{\max}(\mathbf{L}_i) = \frac{1}{n}\sum_{i=1}^{n}L_i \overset{(8)}{\leq} \frac{\nu}{n}L_{\max}. \tag{21}$$

Hence

$$\frac{L}{\mu} + \frac{\mathcal{L}_{\max}}{\mu n} \leq \frac{\nu}{n}\frac{L_{\max}}{\mu} + \frac{2\nu_1}{\sqrt{nd}}\frac{L_{\max}}{\mu} = \mathcal{O}\left(\frac{1}{n}\frac{L_{\max}}{\mu}\right),$$

which guarantees $n$ times fewer communication rounds with the same number of bits per round. In other words, each node communicates $\mathcal{O}(d/n)$ bits to the master in each iteration, which gives us $\mathcal{O}(d)$ communication per communication round. Thus, overall communication complexity to achieve $\varepsilon > 0$ accuracy is

$$\mathcal{O}\left(\frac{d}{n}\frac{L_{\max}}{\mu}\log\frac{1}{\varepsilon}\right).$$

### D.3 Proof of Theorem 4: DIANA+ with block quantization

As already mentioned, for DIANA+ each node aims to minimize $\omega_i + \frac{1}{n\mu}\mathcal{L}(\mathcal{Q}_{h_i}^B, \mathbf{L}_i)$ with respect to its quantization steps $h_i$. Notice that

$$
\begin{aligned}
\omega_i + \frac{1}{n\mu}\mathcal{L}(\mathcal{Q}_{h_i}^B, \mathbf{L}_i) &\leq \max_{l\in[B]} h_{i,l}\sqrt{d_l} + \max_{l\in[B]}\frac{h_{i,l}}{\mu n}\|\mathbf{Diag}(\mathbf{L}_i^{ll})\| \\
&\leq 2\max_{l\in[B]} h_{i,l}\left(\sqrt{d_l} + \frac{1}{\mu n}\|\mathbf{Diag}(\mathbf{L}_i^{ll})\|\right).
\end{aligned}
$$

This leads to the following optimization problem with respect to $h$:

$$
\begin{aligned}
\min_{h\in\mathbb{R}^B} \quad &\max_{1\leq l\leq B} h_l\left(\sqrt{d_l} + \frac{1}{\mu n}\|\mathbf{Diag}(\mathbf{L}_i^{ll})\|\right) \\
\text{s.t.} \quad &\sum_{l=1}^{B}\left(\frac{1}{h_l^2} + \frac{\sqrt{d_l}}{h_l}\right) + B = \beta, \ h_l > 0.
\end{aligned}
\tag{22}
$$

which is solved similar to (11). Denote

$$A_{il} \overset{\text{def}}{=} \sqrt{d_l} + \frac{1}{\mu n}\|\mathbf{Diag}(\mathbf{L}_i^{ll})\|, \quad \widetilde{T}_{iB} \overset{\text{def}}{=} \frac{1}{d}\sum_{l=1}^{B}\sqrt{d_l}A_{il}.$$

Analogous to (11), the solution of (22) has the following form

$$h_{il} = \frac{\widetilde{\delta}_{iB}}{A_{il}}, \ l \in [B],$$

where $\widetilde{\delta}_{iB}$ is determined by the constraint equality of (22) as

$$\widetilde{\delta}_{iB} = \frac{d\widetilde{T}_{i,B}}{2(\beta - B)} + \sqrt{\frac{d^2\widetilde{T}_{i,B}^2}{4(\beta - B)^2} + \frac{d\widetilde{T}_{i,1}^2}{\beta - B}} \leq \frac{d}{\beta - B}\widetilde{T}_{i,B} + \sqrt{\frac{d}{\beta - B}}\widetilde{T}_{i,1}.$$

Let us estimate $\widetilde{T}_{i,1}$ and $\widetilde{T}_{i,n}$ using the assumptions $B = n$ and (for the sake of simplicity) $d_l = d/n$.

$$
\begin{aligned}
\widetilde{T}_{i1} &= \frac{1}{\sqrt{d}}\left(\sqrt{d} + \frac{1}{\mu n}\|\mathbf{Diag}(\mathbf{L}_i)\|\right) = 1 + \frac{1}{\mu n \sqrt{d}}\sum_{j=1}^{d}\mathbf{L}_{i;jj} \le 1 + \frac{\nu_1 L_{\max}}{\mu n \sqrt{d}} \\
\widetilde{T}_{in} &= \frac{1}{d}\sum_{l=1}^{n}\sqrt{\frac{d}{n}}\left(\sqrt{\frac{d}{n}} + \frac{1}{\mu n}\|\mathbf{Diag}(\mathbf{L}_i^{ll})\|\right) = 1 + \frac{1}{\mu n \sqrt{nd}}\sum_{l=1}^{n}\|\mathbf{Diag}(\mathbf{L}_i^{ll})\| \\
&\le 1 + \frac{1}{\mu n \sqrt{nd}}\sum_{j=1}^{d}\mathbf{L}_{i;jj} = 1 + \frac{\nu_1 L_{\max}}{\mu n \sqrt{nd}}.
\end{aligned}
$$

Next, using $\beta = d/n + n$ and $\nu_1 = \mathcal{O}(1)$, we get

$$
\begin{aligned}
\omega_i + \frac{1}{n\mu}\mathcal{L}(\mathcal{Q}_{h_i}^n, \mathbf{L}_i) &\le 2\widetilde{\delta}_{in} \\
&\le \frac{2d}{\beta - n}\widetilde{T}_{in} + 2\sqrt{\frac{d}{\beta - n}}\widetilde{T}_{i1} \\
&= 2n\widetilde{T}_{in} + 2\sqrt{n}\widetilde{T}_{i1} \\
&\le 2n\left(1 + \frac{\nu_1 L_{\max}}{\mu n \sqrt{nd}}\right) + 2\sqrt{n}\left(1 + \frac{\nu_1 L_{\max}}{\mu n \sqrt{d}}\right) = \mathcal{O}\left(n + \frac{1}{\sqrt{nd}}\frac{L_{\max}}{\mu}\right).
\end{aligned}
$$

Together with (21), we complete the proof with the following iteration complexity:

$$
\mathcal{O}\left(n + \frac{1}{n}\frac{L_{\max}}{\mu} + \frac{1}{\sqrt{nd}}\frac{L_{\max}}{\mu}\right).
$$

# E    Proofs for Section 5: Quantization with varying steps

In this part of the appendix we provide missing proofs and detailed arguments of Section 5.

## E.1    An encoding scheme for $\mathcal{Q}_h$ operator

To communicate a vector of the form $\mathcal{Q}_h(x)$, we adapt the encoding scheme of Albasyoni et al. [2020]. From the definition, we have

$$[\mathcal{Q}_h(x)]_j = \|x\| \cdot \text{sign}(x_j \hat{k}_j) \cdot \hat{k}_j h_j$$

for all $j \in [d]$, where $\hat{k}_j \geq 0$ are non-negative random variables coming from (9). Thus, we need to encode the magnitude $\|x\|$, signs $\text{sign}(x_j \hat{k}_j)$ and non-negative integers $\hat{k}_j$.

For the magnitude $\|x\|$ we need just 31 bits. Let $n_0 \overset{\text{def}}{=} |\{j \in [d] \colon \hat{k}_j = 0\}|$ be the number of coordinates $x_j$ that are compressed to 0. To communicate signs $\{\text{sign}(x_j \hat{k}_j) \colon j \in [d]\}$, we first send the locations of those $n_0$ coordinates and then $d - n_0$ bits for the values $\pm 1$. Sending $n_0$ positions can be done by sending $\log d$ bits representing the number $n_0$, followed by $\log \binom{d}{n_0}$ bits for the positions. For the signs, we need $\log d + \log \binom{d}{\hat{n}_0} + d - \hat{n}_0 \leq \log d + d \log 3$ bits at most. Finally, it remains to encode $\hat{k}_j$'s for which we only need to send nonzero entries since the positions of $\hat{k}_j = 0$ are already encoded. We encode $\hat{k}_j \geq 1$ with $\hat{k}_j$ bits: $\hat{k}_j - 1$ ones followed by 0. Hence, the expected number of bits to encode $\hat{k}_j$'s is

$$\mathbb{E}\left[\sum_{j=1}^{d} \hat{k}_j\right] \overset{(9)}{=} \sum_{j=1}^{d} \frac{v_j}{h_j} \leq \sqrt{\sum_{j=1}^{d} v_j^2}\sqrt{\sum_{j=1}^{d} \frac{1}{h_j^2}} = \sqrt{\sum_{j=1}^{d} \frac{1}{h_j^2}} = \|h^{-1}\|,$$

where $v_j = \frac{|x_j|}{\|x\|}$.

In total, $\mathcal{Q}_h(x)$ can be encoded by

$$31 + \log d + \log \binom{d}{\hat{n}_0} + d - \hat{n}_0 + \|h^{-1}\|$$

bits. Lastly, the $\log \binom{d}{\hat{n}_0}$ term can be upper bounded by the binary entropy function $H_2(t) \overset{\text{def}}{=} -t \log t - (1-t) \log(1-t)$ (see [Albasyoni et al., 2020] for more details), and the expected number of encoding bits for $\mathcal{Q}_h(x)$ can be upper bounded by

$$31 + \log d + d H_2\left(\frac{\|\hat{x}\|_0}{d}\right) + \|\hat{x}\|_0 + \|h^{-1}\|,$$

where $\hat{x} = \mathcal{Q}_h(x)$.

## E.2 Proof of the variance bound (14)

Let $v \in \mathbb{R}^d$ be the unit vector with non-negative entries $v_j = |x_j|/\|x\|$ for $j \in [d]$. Then

$$
\begin{aligned}
\mathbb{E}\left[\|\mathcal{Q}_h(x) - x\|_{\mathbf{L}}^2\right] &= \mathbb{E}\left[\left\|\|x\| \cdot \operatorname{sign}(x) \cdot \xi\left(\frac{|x|}{\|x\|}\right) - \|x\| \cdot \operatorname{sign}(x) \cdot \frac{|x|}{\|x\|}\right\|_{\mathbf{L}}^2\right] \\
&= \|x\|^2 \mathbb{E}\left[\|\xi(v) - v\|_{\mathbf{L}}^2\right] \\
&= \|x\|^2 \mathbb{E}\left[\sum_{j,l=1}^{d} \mathbf{L}_{jl}\left(\xi_j(v_j) - v_j\right)\left(\xi_l(v_l) - v_l\right)\right] \\
&= \|x\|^2 \sum_{j=1}^{d} \mathbf{L}_{jj} \mathbb{E}\left[\left(\xi_j(v_j) - v_j\right)^2\right] \qquad (23) \\
&= \|x\|^2 \sum_{j=1}^{d} \mathbf{L}_{jj}\left(v_j - k_j h_j\right)\left(\left(k_j + 1\right)h_j - v_j\right) \\
&= \|x\|^2 \sum_{j=1}^{d} \mathbf{L}_{jj} h_j^2\left(\frac{v_j}{h_j} - k_j\right)\left[1 - \left(\frac{v_j}{h_j} - k_j\right)\right] \\
&\leq \|x\|^2 \sum_{j=1}^{d} \mathbf{L}_{jj} h_j^2 \min\left(1, \frac{v_j}{h_j}\right) \\
&\leq \min\left(\sum_{j=1}^{d} \mathbf{L}_{jj} h_j^2, \sum_{j=1}^{d} \mathbf{L}_{jj} h_j v_j\right)\|x\|^2 \\
&\leq \min\left(\sum_{j=1}^{d} \mathbf{L}_{jj} h_j^2, \sqrt{\sum_{j=1}^{d} \mathbf{L}_{jj}^2 h_j^2}\right)\|x\|^2. \\
&= \min\left(\|\operatorname{\mathbf{Diag}}(\mathbf{L}) h^2\|_1, \|\operatorname{\mathbf{Diag}}(\mathbf{L}) h\|\right)\|x\|^2,
\end{aligned}
$$

which implies (14).

## E.3 Proof of Theorem 5: DCGD+ with varying quantization steps

Based on the upper bound (14) and the communication constraint given by $\|h_i^{-1}\| = \beta$ for some $\beta > 0$, we get the optimization problem

$$
\min_{h_i} \quad \|\operatorname{\mathbf{Diag}}(\mathbf{L}_i) h_i\| \quad \text{subject to} \quad \|h_i^{-1}\| = \beta, \qquad (24)
$$

for choosing the optimal quantization parameters $h_{i;j}$. This problem has a closed form solution. Indeed, due to the KKT conditions, we have

$$
\frac{\mathbf{L}_{i;j}^2 h_{ij}^4}{\sqrt{\sum_{t=1}^{d} \mathbf{L}_{i;t}^2 h_{it}^2}} = 2\zeta, \quad \zeta\left(\sum_{t=1}^{d} h_{ij}^2 - \beta^2\right) = 0,
$$

where $\zeta$ is the multiplier. Solving this leads to the solution:

$$
h_{i;j} = \frac{1}{\beta}\sqrt{\frac{\sum_{t=1}^{d} \mathbf{L}_{i;t}}{\mathbf{L}_{i;j}}}. \qquad (25)
$$

For the solution (25) we have

$$
\begin{aligned}
\widetilde{\mathcal{L}}(\mathcal{Q}_{h_i}, \mathbf{L}_i) &\leq \sqrt{\sum_{j=1}^{d} \mathbf{L}_{i;jj}^2 h_{i;j}^2} = \frac{1}{\beta}\sqrt{\sum_{j=1}^{d} \mathbf{L}_{i;jj}^2 \frac{\sum_{l=1}^{d} \mathbf{L}_{i;ll}}{\mathbf{L}_{i;jj}}} = \frac{1}{\beta}\sum_{j=1}^{d} \mathbf{L}_{i;jj} \\
&\leq \frac{\nu_1}{\beta} \max_{j \in [d]} \mathbf{L}_{i;jj} \leq \frac{\nu_1}{\beta} L_i = \frac{\nu_1}{\beta} L_{\max}. \qquad (26)
\end{aligned}
$$

Therefore, if both parameters $\nu$ and $\nu_2$ are $\mathcal{O}(1)$, then the rate (6) of DCGD+ becomes $\mathcal{O}(\frac{L_{\max}}{n\mu} + \frac{L_{\max}}{\beta n\mu})$. To make a fair comparison against DCGD, we need to fix $\mathcal{O}(\frac{d}{n})$ number of bits each node communicates to the master server. Now, to make DCGD+ communicate the same number of bits, we set $\beta = \mathcal{O}(\frac{d}{n})$. Hence we have the following iteration complexity for DCGD+ based on solution (25):

$$\mathcal{O}\left( \frac{1}{n} \frac{L_{\max}}{\mu} + \frac{1}{d} \frac{L_{\max}}{\mu} \right)$$

which is $\min(n, d)$ times better than the one of DCGD.

### E.4 Proof of Theorem 6: DIANA+ with varying quantization steps

Denote $A_{ij} \overset{\text{def}}{=} \frac{\mathbf{L}_{i;jj}}{n\mu}$. Note that

$$
\begin{aligned}
\omega_i + \frac{\mathcal{L}_i}{n\mu} &\leq \min\left( \sum_{j=1}^{d} h_{i;j}^2, \sqrt{\sum_{j=1}^{d} h_{i;j}^2} \right) + \frac{1}{n\mu} \min\left( \sum_{j=1}^{d} \mathbf{L}_{i;jj} h_{i;j}^2, \sqrt{\sum_{j=1}^{d} \mathbf{L}_{i;jj}^2 h_{i;j}^2} \right) \\
&= \min\left( \sum_{j=1}^{d} h_{i;j}^2, \sqrt{\sum_{j=1}^{d} h_{i;j}^2} \right) + \min\left( \sum_{j=1}^{d} \frac{\mathbf{L}_{i;jj}}{n\mu} h_{i;j}^2, \sqrt{\sum_{j=1}^{d} \left(\frac{\mathbf{L}_{i;jj}}{n\mu}\right)^2 h_{i;j}^2} \right) \\
&\leq \min\left( \sum_{j=1}^{d} h_{i;j}^2 + \sum_{j=1}^{d} \frac{\mathbf{L}_{i;jj}}{n\mu} h_{i;j}^2, \sqrt{\sum_{j=1}^{d} h_{i;j}^2} + \sqrt{\sum_{j=1}^{d} \left(\frac{\mathbf{L}_{i;jj}}{n\mu}\right)^2 h_{i;j}^2} \right) \\
&\leq \min\left( \sum_{j=1}^{d} \left(1 + A_{ij}\right) h_{i;j}^2, \sqrt{2 \sum_{j=1}^{d} \left(1 + A_{ij}^2\right) h_{i;j}^2} \right) \\
&\leq \sum_{j=1}^{d} \left(1 + A_{ij}\right) h_{i;j}^2.
\end{aligned}
$$

We solve the optimization problem

$$\min_{h_i} \quad \sum_{j=1}^{d} \left(1 + A_{ij}\right) h_{i;j}^2 \quad \text{subject to} \quad \left\| h^{-1} \right\| = \beta, \tag{27}$$

which has a closed form solution. Indeed, due to the KKT conditions, we have:

$$h_{i;j} = \frac{1}{\beta} \sqrt{\frac{\sum_{l=1}^{d} \sqrt{1 + A_{il}^2}}{\sqrt{1 + A_{ij}^2}}}. \tag{28}$$

For the solution (28) we have

$$
\begin{aligned}
\omega_i + \frac{\widetilde{\mathcal{L}}_i}{n\mu} &\leq \sqrt{2 \sum_{j=1}^{d} \left(1 + A_{ij}^2\right) h_{i;j}^2} = \frac{\sqrt{2}}{\beta} \sum_{j=1}^{d} \sqrt{1 + A_{ij}^2} = \frac{\sqrt{2}}{\beta} \sum_{j=1}^{d} \left(1 + A_{ij}\right) \\
&= \frac{\sqrt{2}d}{\beta} + \frac{\sqrt{2}}{\beta n\mu} \sum_{j=1}^{d} \mathbf{L}_{i;jj} \leq \frac{\sqrt{2}d}{\beta} + \frac{\sqrt{2}\nu_1}{\beta n} \frac{L_{\max}}{\mu},
\end{aligned}
$$

which further leads to $\mathcal{O}(n + \frac{1}{n} \frac{L_{\max}}{\mu} + \frac{1}{d} \frac{L_{\max}}{\mu})$ iteration complexity if $\nu_1 = \mathcal{O}(1)$ and $\beta = \mathcal{O}(\frac{d}{n})$.

# F   Notation Table

**Table 3:** Notation we use throughout the paper.

| | Basic | |
|---|---|---|
| $d$ | number of the model parameters to be trained | |
| $n$ | number of the nodes/workers in distributed system | |
| $[n]$ | $\overset{\text{def}}{=} \{1, 2, \ldots, n\}$ | |
| $f : \mathbb{R}^d \to \mathbb{R}$ | overall empirical loss/risk | (1) |
| $f_i : \mathbb{R}^d \to \mathbb{R}$ | local loss function associated with data owned by the node $i \in [n]$ | (1) |
| $R : \mathbb{R}^d \to \mathbb{R}$ | (possibly non-smooth) regularization | (1) |
| $x^*$ | trained model, i.e. optimal solution to (1) | |
| $\varepsilon$ | target accuracy | |
| $\|x\|_0$ | $\overset{\text{def}}{=} \#\{j \in [d] : x_j \neq 0\}$, number of nonzero entries of $x \in \mathbb{R}^d$ | |
| $\|x\|$ | $\overset{\text{def}}{=} \sqrt{\sum_{j=1}^d x_j^2}$, the standard Euclidean norm of $x \in \mathbb{R}^d$ | |
| $D_f(x, y)$ | Bregman divergence between $x$ and $y$ with respect to $f$ for $x, y \in \mathbb{R}^d$ | |
| | Standard | |
| $\mu$ | strong convexity parameter of $f$ | Asm. 2 |
| $L$ | smoothness constant of $f$, namely $L = \lambda_{\max}(\mathbf{L})$ | (2) |
| $L_i$ | smoothness constant of $f_i$, namely $L_i = \lambda_{\max}(\mathbf{L}_i)$ | |
| $L_{\max}$ | $\overset{\text{def}}{=} \max_{i \in [n]} L_i$ | |
| $\mathcal{C}$ | (possibly randomized) compression operator $\mathcal{C} : \mathbb{R}^d \to \mathbb{R}^d$ | |
| $\mathbb{B}^d(\omega)$ | class of compressors with $\mathbb{E}[\mathcal{C}(x)] = x$, $\mathbb{E}\left[\|\mathcal{C}(x) - x\|^2\right] \leq \omega\|x\|^2$, $\forall x \in \mathbb{R}^d$ | |
| $\mathcal{C}_i$ | compression operator controlled by node $i$ | |
| $\omega_i$ | variance of compression operator $\mathcal{C}_i$ | |
| $\omega_{\max}$ | $\overset{\text{def}}{=} \max_{i \in [n]} \omega_i$ | |
| $\gamma$ | step-size parameter in DCGD+ and DIANA+ methods | |
| $\alpha$ | learning rate for the local optimal gradients in DIANA+ | |
| | Matrix Smoothness | |
| $\mathbf{L}$ | smoothness matrix of $f$ | (3) |
| $\mathbf{L}^{1/2}$ | square root of symmetric and positive semidefinite matrix $\mathbf{L}$ | |
| $\mathbf{L}^\dagger$ | Moore–Penrose inverse of matrix $\mathbf{L}$ | |
| $\mathbf{L}_i$ | smoothness matrix of $f_i$ | |
| $\mathbf{L}_{i;j}, \mathbf{L}_{i;jj}$ | $j^{th}$ diagonal element of $\mathbf{L}_i$ | |
| $\mathcal{L}(\mathcal{C}, \mathbf{L})$ | $\overset{\text{def}}{=} \inf \left\{ \mathcal{L} \geq 0 : \mathbb{E}\|\mathcal{C}(x) - x\|_{\mathbf{L}}^2 \leq \mathcal{L}\|x\|^2 \ \forall x \in \mathbb{R}^d \right\} \leq \omega\lambda_{\max}(\mathbf{L})$ | |
| $\mathcal{L}_i$ | $\overset{\text{def}}{=} \mathcal{L}(\mathcal{C}_i, \mathbf{L}_i)$ | (4) |
| $\mathcal{L}_{\max}$ | $\overset{\text{def}}{=} \max_{i \in [n]} \mathcal{L}(\mathcal{C}_i, \mathbf{L}_i) = \max_{i \in [n]} \mathcal{L}_i$ | (4) |
| $\nu, \ \nu_1$ | $\nu \overset{\text{def}}{=} \frac{\sum_{i=1}^n L_i}{\max_{i \in [n]} L_i}$ and $\nu_1 \overset{\text{def}}{=} \max_{i \in [n]} \frac{\sum_{j=1}^d \mathbf{L}_{i;j}}{\max_{j \in [d]} \mathbf{L}_{i;j}}$ | Def. 8 |
| | Quantization | |
| $s$ | number of quantization levels | |
| $B$ | number of blocks to divide the space $\mathbb{R}^d$ | |
| $l$ | index for blocks, i.e. $l \in [B]$ | |
| $d_l$ | dimension of the $l^{th}$ subspace in $\mathbb{R}^d$, in particular $\sum_{l=1}^B d_l = d$ | |
| $x^l$ | $l^{th}$ block of coordinates of $x \in \mathbb{R}^d$ | |
| $\mathbf{L}^{ll}$ | $l^{th}$ diagonal block matrix of $\mathbf{L}$ with sizes $d_l \times d_l$ | |
| $h_{i;l}$ | quantization step of $l^{th}$ block for node $i$ | |
| $\beta$ | parameter controlling the number of encoding bits | |
| $j$ | index for coordinates, i.e. $j \in [d]$ | |
| $h_{i;j}$ | quantization step of $j^{th}$ coordinate for node $i$ | (25) |