# OpenReview forum: "Theoretically Better and Numerically Faster Distributed Optimization with Smoothness-Aware Quantization Techniques"
_NeurIPS.cc/2022/Conference — NeurIPS 2022 Accept_

### Official Review · Reviewer_DNd9 · 2022-07-10

**Rating:** 7
**Confidence:** 3
**Soundness:** 3 good
**Presentation:** 2 fair
**Contribution:** 3 good

**Summary:**

This paper considers the problem of distributed optimization over a parameter-server framework wherein the information exchanged from the workers to the server at every iteration is compressed to reduce the communication demand. To design better algorithms for this, the authors consider a different assumption on the smoothness structure of the objective function. They utilize the notion of **matrix smoothness** introduced in *Safaryan et al.* To solve the problem $\text{min}_{\mathbf{x \in \mathbb{R}^d}} f(\mathbf{x})$, where $f(\mathbf{x}) = \frac{1}{n}\Sigma f_i (\mathbf{x})$, existing works assume that each local function $f_i$, as well as the global objective $f$, is $L$-smooth. This assumption ignores the heterogeneity in the structure of the local objective functions, and the authors in this paper use **matrix-smoothness** to exploit this heterogeneity for designing compressors at each worker differently and achieve better convergence rates.

Scalar $L$-smoothness of a function $\phi(\mathbf{x})$ implies that the function curvature in the neighborhood of any point $\mathbf{x} \in \mathbb{R}^d$ in the domain of f can be upper bounded by a quadratic function. In other words, the Bregman divergence with respect to $\phi$ can be upper bounded by the Euclidean distance (with appropriate scaling), i.e.

$D_{\phi}(\mathbf{x}, \mathbf{y}) \leq \frac{L}{2}\lVert \mathbf{x} - \mathbf{y} \rVert_2^2$.

The idea behind matrix $\mathbf{L}$-smoothness for any symmetric positive semi-definite (PSD) matrix $\mathbf{L}$ is to generalize this upper bound to Euclidean distance in a transformed space, i.e.,

$D_{\phi}(\mathbf{x}, \mathbf{y}) \leq \frac{1}{2}\lVert \mathbf{x} - \mathbf{y} \rVert_\mathbf{L}^2$.

Since $\mathbf{L}$ is PSD, its eigenvalue decomposition can be given as $\mathbf{L} = \mathbf{U}\mathbf{\Lambda}\mathbf{U}^T$, and hence for any $\mathbf{z} \in \mathbb{R}^d$,

$\lVert \mathbf{z} \rVert_\mathbf{L}^2 = \mathbf{z}^T \mathbf{L} \mathbf{z} = \mathbf{z}^T \mathbf{U}\mathbf{\Lambda}\mathbf{U}^T \mathbf{z}$, which gives,

$D_{\phi}(\mathbf{x}, \mathbf{y}) \leq \frac{1}{2}\lVert\mathbf{\Lambda}^{1/2}\mathbf{U}^T( \mathbf{x} - \mathbf{y})) \rVert_2^2$.

That is, this gives us a more refined (as well as generalized over scalar $L$-smoothness) information about the smoothness of the objective function, stating that the Bregman divergence of is upper bounded by the Euclidean distance between $\mathbf{x}$ and $\mathbf{y}$ after they have been unitarily transformed into the column space of $\mathbf{U}$, and each coordinate scaled differently in the transformed space. This **matrix smoothness** characterization of the objective function is a generalization of the scalar smoothness characterization, and its usefulness becomes apparent in designing compressors for the gradients of such functions.

From my high-level understanding, this is because of the following reason (I request the authors to correct me if I am mistaken) -- When existing works design compressors for the gradients of functions for which only scalar $L$-smoothness information is known, the usual convergence analysis of the expected suboptimality gap is upper bounded by an expression consisting of the compression error measured in the Euclidean metric, i.e. $\mathbb{E}\lVert C(\mathbf{x}) - \mathbf{x} \rVert_2^2$ . This work considers general unbiased compressors $C(\cdot)$ that satisfy $\mathbb{E}\lVert C(\mathbf{x}) - \mathbf{x} \rVert_2^2 \leq \omega \lVert \mathbf{x} \rVert_2^2$. Here, $\omega \geq 0$ characterizes the expected compression error of $C(\cdot)$. On the other hand, when the matrix smoothness characterization of objective functions is assumed, since the Bregman divergence is upper bounded by a "transformed and scaled" Euclidean distance, in the convergence analysis, the expected suboptimality gap is upper bounded by an expression consisting of the compression error measured in the "transformed and scaled" Euclidean metric, i.e. $\mathbb{E}\lVert C(\mathbf{x}) - \mathbf{x} \rVert_{\mathbf{L}}^2$. This "transformed and scaled" compression error can be characterized similar to before as, $\mathbb{E}\lVert C(\mathbf{x}) - \mathbf{x} \rVert_{\mathbf{L}}^2 \leq \mathcal{L}\lVert \mathbf{x} \rVert_2^2$. Note that:

$\mathbb{E}\lVert C(\mathbf{x}) - \mathbf{x} \rVert_{\mathbf{L}}^2 = \mathbb{E}\lVert \mathbf{\Lambda}^{1/2}\mathbf{U}^T(C(\mathbf{x}) - \mathbf{x}) \rVert_2^2 \leq \lambda_{max}(\mathbf{L})\mathbb{E}\lVert(C(\mathbf{x}) - \mathbf{x}) \rVert_2^2 \leq \lambda_{max}(\mathbf{L})\omega \lVert \mathbf{x} \rVert_2^2$.

So, the matrix smoothness approach yields *clear theoretical benefits* when the "transformed and scaled" upper bound coefficient $\omega \lambda_{max}(\mathbf{L})$ is smaller than the "Euclidean" coefficient, $\omega$. I understand this is *not* the only benefit of the matrix smoothness characterization, but an intuitive understanding nevertheless about why it should be better than the existing approaches.

The authors make use of this fact to propose smoothness-aware distributed methods with general unbiased compressors. They extend existing algorithms -- Distributed Compressed Gradient Descent (DCGD) of *Khirirat et al.* and DIANA *(Mischenko et al.)* and theoretically show benefits in the convergence rate, while also numerically verifying them. They then move on to concrete examples of unbiased compression, specifically non-subtractively dithered quantization. They propose **block quantization** and **quantization with varying steps**, and have solved optimization problems to determine the optimal number of quantization steps for each worker.


**Questions:**

The work is great and has enough theoretical and numerical simulations. However I have several concerns and I would appreciate if the authors addressed them. I would be more than happy to increase my review rating if they are appropriately addressed. They are as follows:

1. The intuition behind the importance of matrix smoothness for compression purposes makes sense. However, in the existing literature, the scalar $L$-smoothness assumption is utilized even in centralized optimization settings in the absence of any compression constraints. Does utilizing the matrix smoothness framework help in improving the convergence rates of uncompressed algorithms in any way? The matrix smoothness assumption was not natural to me after a first reading of this paper, so I resorted to reading the prior work of Safaryan et al. that introduced this notion. Safaryan et al. (https://arxiv.org/pdf/2102.07245.pdf) mention on Page 5 (first paragraph) -- *"Since the stepsizes and convergence rates of first-order methods depend on the smoothness constant(s) employed, convergence analysis relying on such crude approximation may be significantly suboptimal, and the methods too slow when implemented following the theory."* If this is the primary intuition behind the efficacy of matrix smoothness framework, shouldn't it also yield improved convergence rates for centralized optimization settings (without any compression) just because it allows for larger step sizes? The authors are requested to comment on this, and if they agree it is relevant, should probably discuss it somewhere in the paper.

2. In eq. (11) (after line 210), where does the $\beta$ come from? Although it is mentioned in the table of notations as a pre-specified parameter to which quantization is constrained, it should be introduced somewhere in the main paper as well. Moreover, in line (212), should it be $\frac{dT_{i,B}^2}{\beta - B}$ instead of $\frac{dT_{1,B}^2}{\beta - B}$ in the last term of the quadratic expression? Furthermore, even if it is a straightforward calculation, please mention how exactly do you arrive at the solution of (11) (perhaps in the appendix somewhere?)

3. In eq. (13), the first term of the objective function is $h_l \sqrt{d_l}$. This comes from the expression $\omega =  \text{min}\left( \frac{d}{s^2}, \frac{\sqrt{d}}{s} \right) = \text{min}\left( h^2d, h\sqrt{d} \right)$. For $h\sqrt{d}$ to be the minimum, we need $h\sqrt{d} \geq 1 \implies \frac{1}{h^2} \leq d$. Also, the constraint requires: $\Sigma_{l \in [B]} \left( \frac{1}{h_l^2} + \frac{\sqrt{d_l}}{h_l} \right) + B = \beta$. For a homogeneous setting where all local objective functions are the same $\mathbf{L}$-matrix smooth, this constraint becomes, $B \left( \frac{1}{h^2} + \frac{\sqrt{d}}{h} + 1 \right) = \beta \implies \beta \leq B (2d + 1)$. Moreover, since in Thm. 5, it is assumed that $\beta = \mathcal{O}(d/n)$, does this gives us any indication as to the regimes within which the proposed analysis will hold and the algorithm is provably beneficial? A straightforward answer may not be obvious, but since we already know that $n$ and $d$ cannot be too large (from the **Limitations** section owing to the increase in computational complexity) and the analysis further constrains the regimes where is it provably beneficial, the authors should add the discussion regarding this. On a similar note, the statement of Thm. 6 mentions "ignoring negligible term $n$". Is the number of workers ($n$) really negligible -- please justify?

4. In *block quantization*, the vector to be quantized is divided into blocks / subvectors and each block is treated independently, with each block having a (possibly) different quantization resolution. Isn't **quantization with varying steps** just not an extreme case of block quantization where each coordinate is a block by itself? Why does the analysis of *block quantization* simplify to that of *quantization with varying steps* for the choice of parameter $B = d$, where B is the number of blocks? Please justify why a separate analysis is necessary in Sec. 5 for quantization with varying steps.

**Limitations:**

The authors have adequately addresses the major limitations of this work in Sec. **A.3** of the appendix and I appreciate them adding this section, as computational complexity and memory requirements were a concern in my mind right from the beginning when I started reading this paper and got introduced to the notion of matrix smoothness. I imagine any reader in the audience would have similar concerns and hence Sec. A.3 is appreciated. Perhaps the authors could also add a footnote to the appendix somewhere in the main text of the paper. Apart from this, I have raised my other relevant concerns in **Strengths and Weaknesses** and **Questions** sections of the review. Once again, I would be more than happy to increase my scores if my concerns are addressed satisfactorily.

**Strengths And Weaknesses:**

**Originality**: This work heavily relies on the work of *Safaryan et al.* that first proposed the matrix smoothness characterization and demonstrated its utility for random sparsification. Extension to general unbiased compressor is a natural extension to consider. Although the analysis of the proposed method is very similar to *Safaryan et al.*, the contributions are non-trivial. The authors have also clarified this already in Appendices **A.1** and **A.2**, highlighting the differences and the challenges in extending the work of *Safaryan et al.* to general unbiased compressors.

**Quality**: This work is pretty good. The authors have packed in a lot of ideas within 9 pages, which makes the presentation of the paper a little difficult to parse in some portions. Since this work is a generalization of Safaryan et al, some things are often not clear in a first reading. I was not aware of the work of Safaryan et al. and hence, had to go through that paper as well to appreciate this work. Although I understand it is difficult, the readers would appreciate if any paper is as self-contained as possible. Perhaps discuss important ideas like matrix smoothness on which you rely on in the appendix in more detail -- explaining the intuition, etc.  I have also reiterated this point in the *limitations* section of the review. I do have some more technical concerns that I have raised in the **Questions** section of the review.

**Clarity**: Although the work is original and non-trivial, I have several concerns about the presentation of the paper which I have listed below. Mostly they are minor, but I would appreciate if the authors address them.

1. In the abstract, the authors claim, "... our smoothness-aware quantization strategies outperform existing quantization schemes as well as the aforementioned smoothness-aware sparsification strategies with respect to **all relevant success measures** ...". "all relevant success measures" is an unnecessary phrase. The proposed smoothness aware strategies are not very efficient in terms of **computational complexity of preprocessing at the workers** (something the authors note in the Limitations section in Appendix **A.3**) as well as the memory requirements. The server needs to store the smoothness matrices of each worker separately -- for high-dimensional problems and a large number of workers this can be detrimental as it demands a $\mathcal{O}(nd^2)$ memory requirement. Perhaps rephrasing sentence to only include the metrics like *number of iterations*, *total amount of bits communicated*, and *wall-clock time* would be better.

2. In the characterization of unbiased compressors after line 73, $y$ is not required in $\forall x, y \in \mathbb{R}^d$.

3. The authors are sometimes inconsistent with their notations. The class of unbiased compressors is denoted as $\mathbb{B}(\omega)$, but in lines 85 and 118 for example, they add a superscript in $\mathbb{B}^d(\omega)$. The authors are requested to correct such typos.

4. In *Assumption 1*, the authors introduce the smoothness matrices $\mathbf{L}_i$ of the local loss functions $f_i$, and then state the $\mathbf{L}$-matrix smoothness of the global average function $f$. It is not clear if $\mathbf{L}$ is introduced for the first time as the smoothness matrix of the global objective $f$ here. Please mention it if that is the case. If $\mathbf{L}$ can be obtained from the local smoothness matrices $\mathbf{L}_i$, the authors are requested to mention that as well.

5. (I am not sure about this) In line 120, is *overparametrized* the correct term? Shouldn't the term *interpolation* be preferred? I guess a problem does not need to be overparametrized to satisfy $\nabla f_i(\mathbf{x^*}) = 0$ for all $i \in [n]$? Please correct me if I am mistaken.

6. (Line 122) (Just a mild suggestion regarding terminology) " ... we show later that the linear rate of DCGD+ can be much better than one of DCGD..." -- The convergence rate of an optimization algorithm is a function of the learning rate. The authors do not actually compare the linear convergence rate, but rather, the iteration complexity. Although they can be easily derived from one another, the iteration complexity is specified for a particular choice of learning rate (in this case, the maximum allowable learning rate is $\frac{1}{L + \frac{2}{n}\mathcal{L}_{max}}$). On the same note, in Theorem 2, the rate as a function of step-size $\gamma$ is not even mentioned in the statement, just the iteration complexity is mentioned. Please ensure consistency in whether you are comparing the convergence rate or the iteration complexity, and ensure uniformity of statements in Theorems 1 and 2.

7. (Lines 159 onwards) -- **Baseline for DCGD** Please mention the theorem/lemma from which the iteration complexity of DCGD, i.e. the expression $\mathcal{O}\left( \frac{L}{\mu} + \frac{\omega L_{max}}{n \mu} \right)$. Also, the authors mention, " ... optimal level of compression variance $\omega = \mathcal{O}(n)$" -- what is meant by "optimal" here and how is it derived -- please cite appropriately? Furthermore, the estimate of quantization variance $\omega = \text{min}\left( \frac{d}{s^2}, \frac{\sqrt{d}}{s} \right)$ is a consequence of the QSGD scheme of Alistarh et al. and the relevant theorem / lemma of that paper should be cited accordingly.

8. (Line 245) "complicated enough" --> "too complicated"

9. What is $D_f$ in the inequality after the equation in line 591? Is is the Bregman divergence between $x$ and $x^*$ with respect to $f$? Please mention. Could not find it in the table of notations either.

**Significance**: I agree that the work is significant enough for low-to-moderate dimensional optimization problems where the matrix smoothness structure can be exploited. Nevertheless, the domain of communication compression in distributed optimization is often motivated by the communication bottleneck of transmitting high dimensional vectors over a network. Does the proposed framework scale well enough for high dimensional problems?

---

> ### Author Response · Authors · 2022-08-02
> **Other Concerns (Part II)**
>
> $\bullet$ _``(I am not sure about this) In line 120, is overparametrized the correct term? Shouldn't the term interpolation be preferred? I guess a problem does not need to be overparametrized to satisfy $\nabla f\_i(\mathbf{x}^*) = 0$ for all $i\in[n]$? Please correct me if I am mistaken.''_
>
> **Answer**: We are glad to replace it with ``interpolation regime'' since it is more accurate. Please note that it is not a big deal if the interpolation is not satisfied because the convergence rate of DIANA+ does not rely on that.
>
> $\bullet$ _``(Line 122) (Just a mild suggestion regarding terminology) " ... we show later that the linear rate of DCGD+ can be much better than one of DCGD..." -- The convergence rate of an optimization algorithm is a function of the learning rate. The authors do not actually compare the linear convergence rate, but rather, the iteration complexity. Although they can be easily derived from one another, the iteration complexity is specified for a particular choice of learning rate (in this case, the maximum allowable learning rate is $\frac{1}{L+\frac{2}{n}\mathcal{L}\_{\max}}$). On the same note, in Theorem 2, the rate as a function of step size is not even mentioned in the statement, just the iteration complexity is mentioned. Please ensure consistency in whether you are comparing the convergence rate or the iteration complexity, and ensure uniformity of statements in Theorems 1 and 2.''_
>
> **Answer**: To avoid confusion, we replace the ``linear rate'' with iteration complexity.
>
> $\bullet$ _``(Lines 159 onwards) Baseline for DCGD Please mention the theorem/lemma from which the iteration complexity of DCGD, i.e. the expression $\mathcal{O}\left(\frac{L}{\mu} + \frac{\omega L\_{\max}}{n\mu}\right)$.''_
>
> **Answer**: The iteration complexity of DCGD can be obtained as follows. We will add it to the appendix.
>
> The aggregated compressed gradient of DCGD is $g(x) = \frac{1}{n}\sum_{i=1}^n \mathcal{C}\_i(\nabla f\_i(x))$. Considering that $\mathbb{E}\[ \mathcal{C}_i(\nabla f\_i(x) ) \] = \nabla f\_i(x)$, $\mathcal{C}\_i\in\mathbb{B}^d(\omega)$, and the independence among $\mathcal{C}_i$, we have $$ \mathbb{E}\[\\|g(x)-\nabla f(x)\\|^2\] = \frac{1}{n^2}\sum\_{i=1}^n \mathbb{E}\[\\|\mathcal{C}\_i(\nabla f\_i(x)) - \nabla f\_i(x)\\|^2\]\leq \frac{\omega}{n^2}\sum\_{i=1}^n\\|\nabla f\_i(x)\\|^2.$$
>
> The $L_i$-smoothness and the convexity of $f_i$ imply $\\|\nabla f\_i(x) \\|^2 \leq 2\\|\nabla f\_i(x) - \nabla f\_i(x^*)\\|^2 + 2\\|\nabla f\_i(x^*)\\|^2 \leq 4L_i D\_{f\_i}(x,x^*) +  2\\|\nabla f\_i(x^*)\\|^2$, where $D\_{f\_i}(x,x^*)$ is the Bregman divergence between $x$ and $x^*$ with respect to $f\_i$. Then, we have $$ \mathbb{E}\[\\|g(x)-\nabla f(x)\\|^2\] \leq \frac{4\omega L\_{\max}}{n}\frac{1}{n}\sum_{i=1}^n D\_{f\_i}(x,x^*) + \frac{2\omega }{n}\frac{1}{n}\sum_{i=1}^n \\|\nabla f\_i(x^*)\\|^2,\quad L_{\max} := \max_i L_i.$$
>
> Consider that $\mathbb{E}\left[g(x)\right] = \nabla f(x)$, $D\_f(x,x^*) = \frac{1}{n}\sum_{i=1}^n D\_{f\_i}(x,x^*)$, and $\\|\nabla f(x) - \nabla f(x^*)\\|^2 \leq 2L D\_f(x,x^*)$.
> $$ \mathbb{E}\[\\|g(x)-\nabla f(x^*)\\|^2\] =  \mathbb{E}\[\\|g(x)-\nabla f(x)\\|^2\] + \\|\nabla f(x) - \nabla f(x^*)\\|^2 \leq 2\left(L+\frac{2\omega L\_{\max}}{n}\right) D\_f(x,x^*) + \frac{2\omega \sigma\_*^2}{n},$$ where we define $\sigma\_*^2:= \frac{1}{n}\sum\_{i=1}^n \\|\nabla f\_i(x^\*)\\|^2$ ($\sigma\_*^2 = 0$ in the interpolation regime). Applying Theorem 4.1. of [Eduard Gorbunov et. al., A unified theory of SGD: Variance reduction, sampling, quantization and coordinate descent], we can derive the $\mathcal{\widetilde{O}}\left(\frac{L}{\mu} + \frac{\omega L\_{\max}}{n\mu}\right)$ iteration complexity of DCGD shown in our paper.

---

> > ### Author Response · Authors · 2022-08-02
> > **Other Concerns (Part II cont'd.)**
> >
> > $\bullet$ _``(Lines 159 onwards) Also, the authors mention, " ... optimal level of compression variance $\omega = \mathcal{O}(n)$" -- what is meant by "optimal" here and how is it derived -- please cite appropriately?''_
> >
> > **Answer**: In Table 1, we would like to compare our results with the **best total communication complexity** of DCGD and DIANA. Note that we only care about the case $\nu	\gtrapprox 1$ $\Leftrightarrow$ $L\_{\max} \lessapprox n L$.
> >
> > *DCGD*: The iteration complexity of DCGD is $T = \tilde{\mathcal{O}}\left(\frac{L_{\max}}{n\mu} + \frac{\omega L_{\max}}{n\mu}\right)$. When applying standard quantization in Alistarh et al. to DCGD, the amount of bits each node communicates is $b = \mathcal{O}(s^2 + s\sqrt{d}) = \mathcal{O}(\max(s^2, s\sqrt{d})) = \mathcal{O}(\frac{d}{\omega})$ since $\omega = \min\left(\frac{d}{s^2}, \frac{\sqrt{d}}{s}\right) = \frac{d}{\max(s^2, s\sqrt{d})}$. Thus, the total communication complexity of DCGD is $n\cdot T \cdot b = \tilde{\mathcal{O}}\left(\frac{d L_{\max}}{\omega \mu} + \frac{d L_{\max}}{\mu}\right)$. The **optimal** total communication complexity of DCGD is $\tilde{\mathcal{O}}\left(\frac{d L_{\max}}{\mu}\right)$ shown in Table 1, which is attained for any $\omega \geq 1$, including $\omega = \mathcal{O}(n)$.
> >
> > *DIANA*:  The iteration complexity of DIANA is $T = \tilde{\mathcal{O}}\left(\omega + \frac{L_{\max}}{\mu} + \frac{\omega L_{\max}}{n\mu}\right)$. When applying standard quantization in Alistarh et al. to DIANA, the number of bits each node communicates is also $b = \mathcal{O}(\frac{d}{\omega})$. Thus, the total communication complexity of DCGD is $n\cdot T \cdot b = \tilde{\mathcal{O}}\left(nd + \frac{n d L_{\max}}{\omega \mu} + \frac{dL_{\max}}{\mu}\right)$. Thus, the optimal total communication complexity of DIANA is $\tilde{\mathcal{O}}\left(nd + \frac{d L_{\max}}{\mu}\right)$, which is attained when $\omega = \mathcal{O}(n)$.
> >
> > $\bullet$ _``(Lines 159 onwards) Furthermore, the estimate of quantization variance $\omega = \min\left(\frac{d}{s^2},\frac{\sqrt{d}}{s}\right)$ is a consequence of the QSGD scheme of Alistarh et al. and the relevant theorem / lemma of that paper should be cited accordingly.''_
> >
> > **Answer**: We have properly cited the relevant lemma in Alistarh et al. in our updated manuscript.
> >
> > $\bullet$ _``(Line 245) complicated enough --> too complicated''_
> >
> > **Answer**: Fixed. Thanks.
> >
> > $\bullet$ _``What is $D_f$ in the inequality after the equation in line 591? Is it the Bregman divergence between $x$ and $x^*$ with respect to $f$? Please mention. Could not find it in the table of notations either.'_
> >
> > **Answer**: Yes, it is the Bregman divergence. We have mentioned that and added it to the table of notations.
> >
> > $\bullet$ _``Does the proposed framework scale well enough for high dimensional problems?''_
> >
> > **Answer**: We did not analyze numerically the scalability of our framework in high dimensions. However, our theory works for diagonal smoothness matrices as well and scalability issues can be handled.

---

> ### Author Response · Authors · 2022-08-02
> **Other Concerns (Part I)**
>
> $\bullet$ _``Perhaps the authors could also add a footnote to the appendix Sec. A.3 somewhere in the main text of the paper. ''_
>
> **Answer**: We added a pointer to Appendix Sec. A.3 in the caption of Table 1.
>
> $\bullet$ _`` The authors have packed in a lot of ideas within 9 pages, which makes the presentation of the paper a little difficult to parse in some portions. Since this work is a generalization of Safaryan et al, some things are often not clear in the first reading. I was not aware of the work of Safaryan et al. and hence, had to go through that paper as well to appreciate this work. Although I understand it is difficult, the readers would appreciate if any paper is as self-contained as possible. Perhaps discuss important ideas like matrix smoothness on which you rely on in the appendix in more detail -- explaining the intuition, etc.''_
>
> **Answer**: We will add some discussion on the intuition of matrix smoothness and its usefulness in the centralized setting.
>
> $\bullet$ _`` In the abstract, the authors claim, "... our smoothness-aware quantization strategies outperform existing quantization schemes as well as the aforementioned smoothness-aware sparsification strategies with respect to all relevant success measures ...". "all relevant success measures" is an unnecessary phrase.''_
>
> **Answer**: It makes sense. We have replaced the "all relevant success measures" in our paper with ``three evaluation metrics''.
>
> $\bullet$ _``In the characterization of unbiased compressors after line 73, $y$ is not required in $\forall x,y\in\mathbb{R}^d$.''_
>
> **Answer**: Good catch. We have fixed it after line 73 of our updated paper.
>
> $\bullet$ _``The authors are sometimes inconsistent with their notations. The class of unbiased compressors is denoted as $\mathbb{B}(\omega)$, but in lines 85 and 118 for example, they add a superscript in $\mathbb{B}^d(\omega)$. The authors are requested to correct such typos.''_
>
> **Answer**: We have corrected those typos. Thanks.
>
> $\bullet$ _``In Assumption 1, the authors introduce the smoothness matrices $\mathbf{L}\_i$ of the local loss functions $f\_i$, and then state the $\mathbf{L}$-matrix smoothness of the global average function $f$. It is not clear if  $\mathbf{L}$ is introduced for the first time as the smoothness matrix of the global objective $f$ here. Please mention it if that is the case. If $\mathbf{L}$ can be obtained from the local smoothness matrices $\mathbf{L}\_i$, the authors are requested to mention that as well.''_
>
> **Answer**: Sorry for the confusion. Up to Section 3, we use notation $\mathbf{L}$ to denote the smoothness matrix of an arbitrary function (e.g. $\phi$ in Definition 2). However, starting from Section 3, instead of introducing a new notation for the global loss function $f$, we gave the role of the smoothness matrix of $f$ to $\mathbf{L}$ in Assumption 1. Perhaps less confusing and a bit heavier in notation would be to denote it by $\mathbf{L}\_f$. We tried to keep notations as simple as possible and adopted the simpler notation $\mathbf{L}$ for $f$. Regarding the relationship between $\mathbf{L}$ and $\mathbf{L}\_i$, it is easy to check that $\mathbf{L} \preceq \frac{1}{n}\sum_{i=1}^n \mathbf{L}\_i$ since $\nabla^2 f(x) = \frac{1}{n}\sum_{i=1}^n \nabla^2 f_i(x) \preceq \frac{1}{n}\sum_{i=1}^n \mathbf{L}\_i$. We added a note on these aspects in our revision (after Assumption 1).

---

> ### Author Response · Authors · 2022-08-02
> **Concern VI: Is quantization with varying steps a extreme case of block quantization?**
>
> $\bullet$ _``In block quantization, the vector to be quantized is divided into blocks / subvectors and each block is treated independently, with each block having a (possibly) different quantization resolution. Isn't quantization with varying steps just not an extreme case of block quantization where each coordinate is a block by itself? Why does the analysis of block quantization simplify to that of quantization with varying steps for the choice of parameter $B = d$, where B is the number of blocks? Please justify why a separate analysis is necessary in Sec. 5 for quantization with varying steps. ''_
>
> **Answer**: We humbly disagree. The quantization with varying steps does not coincide with the extreme case of block quantization when $B=d$. First of all, typically one encodes the number of bits after applying block quantization to send the $B$ norms of the blocks, which does not save in communication if we set $B=d$. Besides, if we treat each coordinate as a block (i.e., $B=d$), the compressed value of the $j$-th coordinate is $\[\mathcal{Q}\_h^d(x)\]\_{j} = |x\_j| \cdot \textrm{sign}(x\_j) \cdot \xi\_j\(\frac{|x\_j|}{|x\_j|}\) = x\_j \cdot \xi\_j\(1\)$, where $\mathbf{E}[\xi\_j\(1\)]=1$. In contrast, the compressed value of the $j$-th coordinate by quantization with varying steps is $\[\mathcal{Q}\_h(x)\]\_j = \|x\| \cdot \textrm{sign}(x\_j) \cdot \xi\_j\(\frac{|x\_j|}{\|x\|}\)$.

---

> ### Author Response · Authors · 2022-08-02
> **Concern V: Is $n$ negligible?**
>
> $\bullet$ _``On a similar note, the statement of Thm. 6 mentions "ignoring negligible term $n$". Is the number of workers ($n$) really negligible -- please justify? ''_
>
> **Answer**: The term $n$ in iteration complexities is negligible because in typical training problems the condition number $\frac{L\_{\max}}{\mu}$ of the problem is significantly bigger (i.e. ill-conditioned problems) than the number of devices $n$.

---

> ### Author Response · Authors · 2022-08-02
> **Concern IV: Homogeneous setting**
>
> $\bullet$ _``In eq. (13), the first term of the objective function is $h_l\sqrt{d_l}$. This comes from the expression $\omega = \min\left(\frac{d}{s^2},\frac{\sqrt{d}}{s}\right) = \min(h^2 d, h\sqrt{d})$. For $h\sqrt{d}$ to be the minimum, we need $h\sqrt{d}\geq 1\Rightarrow \frac{1}{h^2}\leq d$. Also, the constraint requires: $\Sigma\_{l \in [B]} \left( \frac{1}{h\_l^2} + \frac{\sqrt{d\_l}}{h\_l} \right) + B = \beta$. For a homogeneous setting where all local objective functions are the same $\mathbf{L}$-matrix smooth, this constraint becomes, $B \left( \frac{1}{h^2} + \frac{\sqrt{d}}{h} + 1 \right) = \beta \implies \beta \leq B (2d + 1)$. Moreover, since in Thm. 5, it is assumed that $\beta = \mathcal{O}(d/n)$. does this give us any indication as to the regimes within which the proposed analysis will hold and the algorithm is provably beneficial? A straightforward answer may not be obvious, but since we already know that $n$ and $d$ cannot be too large (from the Limitations section owing to the increase in computational complexity) and the analysis further constrains the regimes where is it provably beneficial, the authors should add the discussion regarding this.''_
>
> **Answer**: For a homogeneous setting where all local objective functions are the same, namely $f\_i(x)=f(x)$ and $\mathbf{L}\_i = \mathbf{L}$, then the constraint $\Sigma_{l \in [B]} \left( \frac{1}{h\_l^2} + \frac{\sqrt{d_l}}{h\_l} \right) + B = \beta$ does not become $B \left( \frac{1}{h^2} + \frac{\sqrt{d}}{h} + 1 \right) = \beta$. Notice that each block $l\in[B]$ has its own quantization step $h_l$ even when all smoothness matrices are the same. Could you please reformulate your question?

---

> ### Author Response · Authors · 2022-08-02
> **Concern III: How do we arrive at the solution of (11)?**
>
> $\bullet$ _``Moreover, in line (212), should it be $\frac{dT\_{i,B}^2}{\beta-B}$ instead of $\frac{dT\_{1,B}^2}{\beta-B}$ in the last term of the quadratic expression? Furthermore, even if it is a straightforward calculation, please mention how exactly do you arrive at the solution of (11) (perhaps in the appendix somewhere?)''_
>
> **Answer**: Thank you for the typo. That expression in line (212) should be $\frac{dT\_{i,1}^2}{\beta-B}$, where $T\_{i,1}:=\frac{1}{d}\sqrt{d}  \\|\textrm{Diag}(\mathbf{L}\_i)\\|$.
>
> For the type of problem (with some $a\_l, c>0$) $$\min\_{h\in\mathbb{R}^B} \max\_{l\in[B]} a\_l h\_l,\quad \textrm{s.t.}\~h_l\geq 0,\~\sum_{l=1}^B \textrm{poly}\left(\frac{1}{h_l}\right) = c,$$ where the polynomial coefficients are non-negative. The optimum is attained when $a\_lh\_l$ is uniform over $l$, i.e., $h_l = \frac{\delta}{a_l}$ ($\delta>0$) and $\delta$ makes the constraint hold. That is the reason why we set $h\_{i,l} = \frac{\delta\_{i,B}}{\\|\textrm{Diag}(\mathbf{L}\_i^{ll})\\|}$ for (11) in our paper. So the rest of the work is to solve $\delta\_{i,B}$ to make the constraint hold. After plugging in $h\_{i,l} = \frac{\delta\_{i,B}}{\\|\textrm{Diag}(\mathbf{L}\_i^{ll})\\|}$, the constraint becomes $ \frac{\sum\_{l=1}^B\\|\textrm{Diag}(\mathbf{L}\_i^{ll})\\|^2}{\delta\_{i,B}^2} + \frac{\sum\_{l=1}^B\sqrt{d\_l}\\|\textrm{Diag}(\mathbf{L}\_i^{ll})\\|}{\delta\_{i,B}} = \beta - B.$ which is equivalent to $\delta\_{i,B}^2 -  \delta\_{i,B}\frac{d}{\beta-B} \frac{1}{d} \sum\_{l=1}^B\sqrt{d\_l}\\|\textrm{Diag}(\mathbf{L}\_i^{ll})\\| - \frac{d}{\beta-B}\frac{1}{d}\sum\_{l=1}^B\\|\textrm{Diag}(\mathbf{L}_i^{ll})\\|^2.$
>
> Its solution can be easily computed by the quadratic formula. Note that we define $T\_{i,B}:= \frac{1}{d} \sum\_{l=1}^B\sqrt{d\_l}\\|\textrm{Diag}(\mathbf{L}\_i^{ll})\\|$ and $T\_{i,1}^2:=\left(\frac{1}{d}\sqrt{d}  \\|\textrm{Diag}(\mathbf{L}\_i)\\|\right)^2 = \frac{1}{d}\sum\_{l=1}^B\\|\textrm{Diag}(\mathbf{L}_i^{ll})\\|^2$. This is how we arrive at the solution of (11).

---

> ### Author Response · Authors · 2022-08-02
> **Concern II: What is beta?**
>
> $\bullet$ _``In eq. (11) (after line 210), where does the $\beta$ come from? Although it is mentioned in the table of notations as a pre-specified parameter to which quantization is constrained, it should be introduced somewhere in the main paper as well.''_
>
> **Answer**: $\beta$ is the ``budget'' of communication that was introduced in (11) to fix the number of bits communicated by the worker. With larger $\beta$, the quantization levels are finer, and hence the algorithms need fewer #iterations to converge at the cost of higher communication cost per iteration.

---

> ### Author Response · Authors · 2022-08-02
> **Concern I: Matrix smoothness for centralized optimization settings?**
>
> $\bullet$ _``The intuition behind the importance of matrix smoothness for compression purposes makes sense. However, in the existing literature, the scalar $L$-smoothness assumption is utilized even in centralized optimization settings in the absence of any compression constraints. Does utilizing the matrix smoothness framework help in improving the convergence rates of uncompressed algorithms in any way? The matrix smoothness assumption was not natural to me after a first reading of this paper, so I resorted to reading the prior work of Safaryan et al. that introduced this notion. Safaryan et al. (https://arxiv.org/pdf/2102.07245.pdf) mention on Page 5 (first paragraph) -- "Since the stepsizes and convergence rates of first-order methods depend on the smoothness constant(s) employed, convergence analysis relying on such crude approximation may be significantly suboptimal, and the methods too slow when implemented following the theory." If this is the primary intuition behind the efficacy of matrix smoothness framework, shouldn't it also yield improved convergence rates for centralized optimization settings (without any compression) just because it allows for larger step sizes? The authors are requested to comment on this, and if they agree it is relevant, should probably discuss it somewhere in the paper.''_
>
> **Answer**: This is a good question and an opportunity for us to further discuss the usefulness of smoothness matrices in the paper. In fact, smoothness matrices have been studied in the randomized coordinate descent literature for several years [Zheng Qu and Peter Richtárik, Coordinate descent with arbitrary sampling II: expected separable overapproximation] [Zheng Qu and Peter Richtárik, Coordinate descent with arbitrary sampling I: algorithms and complexity] [Peter Richtárik and Martin Takáč, On optimal probabilities in stochastic coordinate descent methods] [Mher Safaryan, Filip Hanzely and Peter Richtárik, Smoothness matrices beat smoothness constants: better communication compression techniques for distributed optimization, Section B].
>
> For example, the 'NSync algorithm of [Peter Richtárik and Martin Takáč, On optimal probabilities in stochastic coordinate descent methods] uses the smoothness matrix to estimate smaller, so-called, ESO parameters $v\_i$ for each coordinate, which then leads to larger stepsizes for the update rule and improved complexity for the algorithm. Note that randomized coordinate descent can be viewed as compressed gradient descent with random sparsification (number of workers $n=1$).

---

> ### Author Response · Authors · 2022-08-02
> **Thank you**
>
> We thank you for the detailed feedback and constructive suggestions!

---

> > ### Comment · Reviewer_DNd9 · 2022-08-09
> > **Acknowledgement of the rebuttal**
> >
> > Dear authors,
> >
> > I thank you for your detailed response. Most of my concerns were clarified. I am leaning towards increasing my score and will do that after the reviewer discussion period.
> >
> > Looking forward to the reviewer discussion period.
> >
> > Best,
> > Reviewer DNd9

---

> > > ### Author Response · Authors · 2022-08-09
> > > **Thanks!**
> > >
> > > Dear Reviewer DNd9,
> > >
> > > Thank you for this, we are most happy to hear this!
> > >
> > > Best regards,
> > >
> > > authors

---

### Official Review · Reviewer_SKPB · 2022-07-11

**Rating:** 7
**Confidence:** 2
**Soundness:** 3 good
**Presentation:** 3 good
**Contribution:** 3 good

**Summary:**

This paper is concerning compressed communication in distributed machine learning.
Smoothness Matrices by Safaryan et al. make it possible to use compressors with different smoothness information for each node, but the choice of compression function is restricted. This paper proposes a generalized method that allows Smoothness Matrices to be applied to any unbiased compressor below a certain variance. This paper further derives how to improve the existing methods called DCGD and DIANA with the proposed framework.

## Afer author feedback
Considering the answers to my questions and other reviewers' question, I raise my score +1 upward.
I expect the discussion about the assumption viloations to be inculded somewhere in the appendix.


**Questions:**

* Is the idea of multiplying L^{1/2} before and after Compressor a basic common practice? Has it been used in other papers? Or is it first conceived for this paper?

* How strictly must assumptions 1 and 2 be followed when dealing with uncontrolled real data?　Which assumption is more likely to cause practical problems in distributed compression learning when violated?


**Limitations:**

Discussed in aeepndx.

**Strengths And Weaknesses:**

## Originality:


The main result extends the new distributed compression learning results proposed last year.

This paper proposes an algorithm that applies an extended version of Smoothenss Matrices to a broader class of compression (quantization) algorithms, though the reviewer cannot fully understand the technical originality of the main result (the first point in Appendix A.2?).
If I correctly understand, the quantization methods applied to DCGD and DIANA are not new.

## Quality:

The validity of the proofs has not been verified.
The technical ingenuity and the main idea are well understood.
The correspondence between the claims of the paper and the supporting theorem and experimental results is clear.



## Clarity:

The structure of the paper and its arguments are written in an easy-to-understand manner.
Even a non-expert reviewer (=me) on this topic can understand the technical innovations and results.


## Significance:

The reviewer mainly works on neural data (image) compression models and is not an expert on compression "communication" protocols in distributed machine learning.
Thus the reviewer is not confident in accurately evaluating this study in the context of the distributed ML community.

However, it is understandable that choosing an appropriate compression model is important to improve the efficiency of distributed compression learning.
The original Smoothness Matrices method is restricted to a certain type of compressor, thus the contribution of the paper seems important.

The numerical experimental results clearly show advancements brought by the proposed framework. However, the reviewer cannot judge how significant these advancements are.

---

> ### Author Response · Authors · 2022-08-02
> **Concern III: Regarding the assumptions**
>
> $\bullet$ _``How strictly must assumptions 1 and 2 be followed when dealing with uncontrolled real data? Which assumption is more likely to cause practical problems in distributed compression learning when violated?''_
>
> **Answer**: For generalized linear models (GLM) (e.g., linear/logistic regression, SVM with smooth hinge loss) with $\ell_2$-regularization, assumptions 1 and 2 can be shown to strictly hold and the analytic expression of $\mathbf{L}\_i$ can be given. For example, $\mathbf{L}\_i = \frac{1}{4m_i}\sum\_{m=1}^{m_i}\mathbf{A}\_{im}^\top\mathbf{A}\_{im} + \lambda$ for $\ell_2$-regularized logistic regression, where $\{\mathbf{A}\_{im} \colon m=1,\dots,m_i\}$ is the real (local) data of device $i$ and $\lambda$ is the regularization parameter.
>
> Beyond GLMs, assumptions 1 and 2 might not hold or the assumptions hold but the analytic expression of $\mathbf{L}_i$ is difficult to obtain. In the former case (e.g. the problem is non-convex), one might consider another algorithm to solve the problem (e.g. MARINA [Gorbunov et al. [2021]]). In the latter case, one possibility is to treat $\mathbf{L}_i$'s as hyper-parameters and learn some *rough approximations of the smoothness matrices from the first order information obtained by running a gradient type method*. This can be done initially as a preprocessing step, after which the matrices are considered ``learned'', and then our compression can be built and used. But we do not have theoretical guarantees for these workarounds.

---

> > ### Comment · Reviewer_SKPB · 2022-08-06
> > **Thanks!**
> >
> > Thank you for your explanation!
> > The latter case fix-up, rough approximation by first order information, sounds interesting and practaical. I understand the approximation loses the theoretical advantage, but may expand the practical applications.

---

> ### Author Response · Authors · 2022-08-02
> **Concern II: Is the idea a common practice?**
>
> $\bullet$ _``Is the idea of multiplying $\mathbf{L}^{1/2}$ before and after Compressor a basic common practice? Has it been used in other papers? Or is it first conceived for this paper?'_
>
> **Answer**: To the best of our knowledge, the idea is not used by other papers in the literature except for Safaryan et al. [2021]. Thus, it is certainly not a basic common practice (yet). The discovery that smoothness information can speed up distributed training is very recent. The difference is that the theory of Safaryan et al. [2021] only covers the random sparsification while ours applies to arbitrary unbiased estimators including the random sparsification and the new quantization methods in this paper.

---

> ### Author Response · Authors · 2022-08-02
> **Concern I: Quantization methods are not new**
>
> $\bullet$ _``If I correctly understand, the quantization methods applied to DCGD and DIANA are not new.''_
>
> **Answer**: Block quantization as a compression scheme was known (see e.g., [Shuai Zheng, Ziyue Huang, James T. Kwok, Communication-Efficient Distributed Blockwise Momentum SGD with Error-Feedback] or [Konstantin Mishchenko, Filip Hanzely, Peter Richtárik, 99% of Distributed Optimization is a Waste of Time: The Issue and How to Fix it]). Quantization with varying steps (introduced in section 5) is new. Both schemes were not analyzed under matrix smoothness assumption though. The varying steps per block or per coordinate are necessary for those quantization methods to derive improved communication complexities by utilizing the matrix smoothness. Both the closed-form expressions of the varying steps and the improved communication complexities are new.

---

> > ### Comment · Reviewer_SKPB · 2022-08-06
> > **Thank you the clarification**
> >
> > So the part of the quantization is newly taylored, Got it!

---

> ### Author Response · Authors · 2022-08-02
> **Thank you**
>
> We thank you for the review!

---

### Official Review · Reviewer_dhtX · 2022-07-11

**Rating:** 6
**Confidence:** 4
**Soundness:** 3 good
**Presentation:** 3 good
**Contribution:** 3 good

**Summary:**

This paper generalizes the smoothness-aware compression strategy to arbitrary unbiased compression. This trick was applied to two existing algorithms, DCGD and DIANA, improving communication complexity. Because it can be applied to any unbiased compression scheme, the quantization was improved further with block quantization and varying steps.

**Questions:**

How does the algorithm perform if the smoothness matrix $L$ is unknown (e.g., it takes too long to find it)? If the matrix is wrong, what will happen to the algorithm?
+ Does the transfer of the matrix $L$ use compression? The matrix $L$ is not sparse, and the size is $d\times d$, which can take a long time.
+ The convergence results require $\nu,$ $\nu_1$ being $O(1)$. In fact, the numbers can be in the order of $n$ and $d$ for IID data. In the numerical experiments, the data is split after random shuffled. So the number for $\nu$ and $\nu_1$ may be too big. It would be interesting to see the performance on heterogeneous data.

**Limitations:**

Majors limitations are already listed in the Appendix A.3.

**Strengths And Weaknesses:**

Strengths
+ The proposed trick uses local second-order information to compress the gradient, largely improving existing compression techniques' performance.
+ The convergence is provided for strongly convex functions under some assumptions.

Weakness
+ The algorithm requires the smoothness matrix $L$, which may not be easily obtained.

---

> ### Author Response · Authors · 2022-08-02
> **Concern IV: Data split**
>
> $\bullet$ _``The convergence results require $\nu$, $\nu_1$ being $O(1)$. In fact, the numbers can be in the order of $n$ and $d$ for IID data. In the numerical experiments, the data is split after random shuffled. So the number for $\nu$ and $\nu_1$ may be too big. It would be interesting to see the performance on heterogeneous data.''_
>
> **Answer**: We made a mistake in our previous manuscript. In our experiments, the data points are actually sorted based on their norms before allocating to local workers, instead of randomly shuffled, to ensure that the data split is heterogeneous. We have fixed it in lines 293-294 of our updated manuscript.

---

> ### Author Response · Authors · 2022-08-02
> **Concern III: Transfer of the smoothness matrix**
>
> $\bullet$ _``Does the transfer of the matrix $\mathbf{L}$ use compression? The matrix $\mathbf{L}$ is not sparse, and the size is $d\times d$, which can take a long time.''_
>
> **Answer**: We did not analyze the compression of the smoothness matrix before communication as it is transferred **only once** before the training begins. Besides, we showed in our experiments that the overhead in communication cost is negligible when the number of iterations is large (the transmitted megabytes do not start from 0 in our plots).
>
> However, in practice, compressing the matrix $\mathbf{L}$ is a good idea. One option for that is to initially estimate a diagonal smoothness matrix that is as easy to communicate (still **only once**) as one full precision gradient. Another option is to directly apply compression to the matrix $\mathbf{L}$ so that the compressed matrix is an over-approximation. For example, let $\mathbf{L} = \sum_{k=1}^d \lambda_k u_ku_k^\top$ be the eigendecomposition of $\mathbf{L}$, where $\lambda_k$ is the $k^{th}$ largest eigenvalue corresponding to eigenvector $u_k$. Then $\mathbf{L} \preceq \sum_{k=1}^r \lambda_k u_ku_k^\top + \sum_{k=r+1}^d \lambda_{r+1} u_ku_k^\top = \sum_{k=1}^r (\lambda_k - \lambda_{r+1}) u_ku_k^\top + \lambda_{r+1}\mathbf{I}$. The latter over-approximation (which serves as a smoothness matrix for $f$) can be transferred with $r d + 1$ floats where $r$ can be chosen small.

---

> ### Author Response · Authors · 2022-08-02
> **Concern II: What if the smoothness matrix is wrong?**
>
> $\bullet$ _``If the matrix is wrong, what will happen to the algorithm?''_
>
> **Answer**: Notice that if $\mathbf{L}$ is a smoothness matrix (could be dense, diagonal or any structure) of $f$, then any matrix $\widetilde{\mathbf{L}} \succeq \mathbf{L}$ is also a smoothness matrix for the same loss function since $\nabla^2 f(x) \preceq \mathbf{L} \preceq \widetilde{\mathbf{L}}$. This implies that the algorithm would still work if the smoothness matrix is over-approximated. However, it is also worth mentioning that the tighter the approximation of smoothness information the better performance of the algorithm (theoretical step sizes depend on smoothness matrices).

---

> > ### Comment · Reviewer_dhtX · 2022-08-06
> > **Thanks for clarifying this**
> >
> > So if the matrix is unknown, it is better to overestimate it than underestimate it because an overestimation still works under the theory.

---

> ### Author Response · Authors · 2022-08-02
> **Concern I: What if the smoothness matrix is unknown?**
>
> $\bullet$ _``How does the algorithm perform if the smoothness matrix  is unknown (e.g., it takes too long to find it)?''_
>
> **Answer**: Note that some information about the smoothness of loss functions must be known, otherwise our framework cannot be applied. Basically, our work is about taking advantage of that information to speed up the training. We agree that estimating dense smoothness matrix $\mathbf{L}$ (i.e., $\nabla^2 f(x) \preceq \mathbf{L}$) could be hard for non-GLM problems because of the $d^2$ number of entries and lack of closed-form expression. However, estimating sparse, such as diagonal, smoothness matrix $\textbf{Diag}(L^1, L^2, \dots,L^d)$ (i.e., $\nabla^2 f(x) \preceq \textbf{Diag}(L^1, L^2, \dots,L^d)$) should be feasible. In fact, all theoretical results and improvement factors we proved are valid for the case of diagonal smoothness matrices as well (notice that both $\nu$ and $\nu_1$ parameters can still be $O(1)$ in this case).

---

> ### Author Response · Authors · 2022-08-02
> **Thank you**
>
> We thank you for the feedback!

---

### Official Review · Reviewer_WNkz · 2022-07-11

**Rating:** 6
**Confidence:** 3
**Soundness:** 3 good
**Presentation:** 4 excellent
**Contribution:** 3 good

**Summary:**

The paper extends Safaryan's smoothness-aware compression method to the general unbiased compression method. Meanwhile, the authors propose two non-linear compressors for DCGD+ and DIANA+.


**Questions:**

Could you give the code for reproduction?

**Limitations:**

The authors have discussed the limitation in the detail.

**Strengths And Weaknesses:**

Strengths

1.This paper has a solid theoretical analysis of convergence and communication complexity.

2.Two non-linear compressors are proposed for DCGD+ and DIANA+ and the theoretical results show their good performance.

3.The authors conduct serious experiments to evaluate their method. They are detailed and easy to follow.


 Weaknesses

1.This paper is an extended paper to Safaryan's, which reduces the significance.

2.Though authors have discussed the technical contribution, the main difficulty of extending to the arbitrary unbiased compression operators seems to be easy to handle.

---

> ### Author Response · Authors · 2022-08-02
> **Concern II: Reproduction**
>
> $\bullet$ _``Could you give the code for reproduction?''_
>
> **Answer**: Sure. The code is provided anonymously at https://anonymous.4open.science/r/Paper_9107_code-1CC6.

---

> ### Author Response · Authors · 2022-08-02
> **Concern I: Significance and technical contributions**
>
> $\bullet$ _``This paper is an extended paper to Safaryan's, which reduces the significance. Though authors have discussed the technical contribution, the main difficulty of extending to the arbitrary unbiased compression operators seems to be easy to handle.''_
>
>
> **Answer**: One reason it might seem easy to handle is how we presented the technical contributions. Nevertheless, we made several non-trivial contributions and outlined limitations we did not resolve in this work. We believe that such an easy-to-follow presentation of technical details would foster future extensions and improvements in smoothness-aware distributed training.

---

> ### Author Response · Authors · 2022-08-02
> **Thank you**
>
> Thank you for the review!

---

### Author Response · Authors · 2022-08-03
**To all reviewers**

Dear reviewers,

Thank you for the comments! We have updated our manuscript and provided detailed responses to address your concerns.

Authors

---

### Author Response · Authors · 2022-08-07
**To all reviewers: we would love to hear from you**

Thanks to all reviewers for your reviews, and thanks to all who managed to find time in their busy schedules (or perhaps even during vacation) to engage with our rebuttal as well!

**Please can you let us know whether there any issues you noticed we did not managed to address? If all issues are addressed, please would you consider adjusting your score? We believe our work is an important theoretical and algorithmic contribution to the fast growing field of communication-efficient distributed learning, and as such we are of the view that it deserves better scores than mere weak accept. We believe our work is timely, and offers substantial results.**

- Reviewer WNkz: Please can you let us know what you think about our response? Our response was brief, since we thought only minor clarification questions were asked. But if we can explain any more parts of our work, let us know.

- Reviewer dhtX: Please can you let us know what you think about our response? Our response was brief, since we thought only minor clarification questions were asked. But if we can explain any more parts of our work, let us know. We notice one comment from you, thanks! But what do you think about our other pieces of author feedback?

- Reviewer SKPB: Thanks for you comments on our rebuttal. It seems to us that you are satisfied with our rebuttal. In case you are indeed satisfied, and if you think this is appropriate, please do consider increasing your score. We think our work is of a significantly higher value than what a "weak accept" seems to indicate. Of course, we are biased. But if you agree, we would love to see this being reflected in the score. Apologies for mentioning this explicitly.

- Reviewer DNd9: Thanks again for the substantial review! Much much appreciated! We would be happy to hear your views on our author response, which is substantial, too, and detailed. Do any issues remain?


Thanks to all!

Authors of "Theoretically Better and Numerically Faster Distributed Optimization with Smoothness-Aware Quantization Techniques"

---

### Meta-Review · Area_Chair_wErG · 2022-08-20

**Recommendation:** Accept
**Confidence:** Certain

**Metareview:**

The reviewers are all in agreement that the paper passes the bar for acceptance, and no significant concerns remain following the review period.  Quite a few questions and concerns were raised in one of the original reviews, and the authors gave very detailed responses that the reviewer found to be sufficiently convincing.  The authors should carefully modify the paper in accordance with these comments/questions (and those of the other reviewers), particularly those of a technical nature.

**Award:**

No

---

### Decision · Program_Chairs · 2022-09-14

Accept